

# Charting the space of ground states with tensor networks

Marvin Qi[1,2], David T. Stephen[1,2,3], Xueda Wen[1,2,4], Daniel Spiegel[1,2,5],
Markus J. Pflaum[2,5], Agnès Beaudry[5] and Michael Hermele[1,2]

**1** Department of Physics, University of Colorado, Boulder, CO 80309, USA
**2** Center for Theory of Quantum Matter, University of Colorado, Boulder, CO 80309, USA
**3** Department of Physics and Institute for Quantum Information and Matter,
California Institute of Technology, Pasadena, California 91125, USA
**4** Department of Physics, Harvard University, Cambridge MA 02138, USA
**5** Department of Mathematics, University of Colorado, Boulder, CO 80309, USA

## Abstract

We employ matrix product states (MPS) and tensor networks to study topological properties of the space of ground states of gapped many-body systems. We focus on families of states in one spatial dimension, where each state can be represented as an injective MPS of finite bond dimension. Such states are short-range entangled ground states of gapped local Hamiltonians. To such parametrized families over $X$ we associate a gerbe, which generalizes the line bundle of ground states in zero-dimensional families (*i.e.* in few-body quantum mechanics). The nontriviality of the gerbe is measured by a class in $H^3(X, \mathbb{Z})$, which is believed to classify one-dimensional parametrized systems. We show that when the gerbe is nontrivial, there is an obstruction to representing the family of ground states with an MPS tensor that is continuous everywhere on $X$. We illustrate our construction with two examples of nontrivial parametrized systems over $X = S^3$ and $X = \mathbb{R}P^2 \times S^1$. Finally, we sketch using tensor network methods how the construction extends to higher dimensional parametrized systems, with an example of a two-dimensional parametrized system that gives rise to a nontrivial 2-gerbe over $X = S^4$.

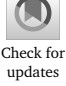

# 1 Introduction and overview

## 1.1 Introduction

The space of gapped quantum many-body systems has a rich topological structure. For instance, suppose we consider a space $\mathfrak{GH}$ of gapped local Hamiltonians in some fixed spatial dimension $d$.[1] It is a familiar idea that distinct gapped phases of matter correspond to the connected components of $\mathfrak{GH}$; each connected component is the (infinite-dimensional) parameter space of a gapped phase. It is less familiar that the topology of the connected components themselves can be non-trivial, as measured for instance via their homotopy groups $\pi_n$. Such non-trivial topology can put constraints on phase diagrams, and lead to interesting phenomena in systems with one or more continuously tuneable parameters.

While our remarks so far emphasize Hamiltonians, we can just as well focus on ground states instead. Let $\mathscr{Q}$ be the space of ground states of gapped local Hamiltonians in the same spatial dimension. Clearly there is a map $\mathfrak{GH} \to \mathscr{Q}$ that sends a Hamiltonian to its ground state. This map is believed to be a homotopy equivalence.[2] Therefore, each connected component of $\mathscr{Q}$ is the space of ground states associated to a phase, and all the homotopy-theoretic properties of this space are the same as the phase's parameter space (connected component of $\mathfrak{GH}$).

The foundational example of this physics lies in few-body quantum mechanics ($d = 0$), and is simply a spin-1/2 particle in a Zeeman field [1,2]

$$H(\vec{B}) = \vec{B} \cdot \vec{\sigma} = B_x \sigma^x + B_y \sigma^y + B_z \sigma^z. \tag{1}$$

---

[1]We also require that Hamiltonians in $\mathfrak{GH}$ have a unique ground state in infinite space $\mathbb{R}^d$. This removes any locii of first-order phase transitions from $\mathfrak{GH}$. It should be noted that this requirement does not remove topologically ordered systems from $\mathfrak{GH}$; such systems only have a ground state degeneracy when space is periodic.

[2]This is explained in a talk of Kitaev [26] and Sec. 2.1.1 of Ref. [24].

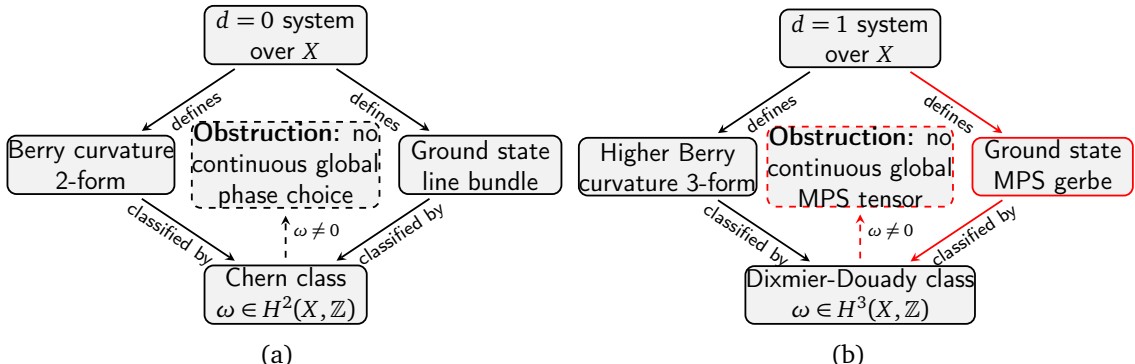

Figure 1: Illustration of approaches to probe the non-trivial topology of the space of ground states in spatial dimension $d = 0$ (a) and $d = 1$ (b). This paper introduces the ground state MPS gerbe and explains why a non-trivial Dixmier-Douady class is an obstruction to finding a continuous MPS tensor defined globally over $X$.

We think of Equation (1) as defining a family of Hamiltonians continuously parametrized by $\vec{B}$; clearly, the spectrum is gapped as long as $\vec{B} \neq 0$. For the discussion below, it is convenient to fix the magnitude $|\vec{B}| = B$, which defines an $S^2$ subspace of parameter space. It is convenient to refer to this example as a *system over $S^2$*, which means we have a family of Hamiltonians (or ground states) with parameters lying in $S^2$; this terminology is defined more generally and precisely below in Sec. 1.2.

The topological non-triviality of the above family of Hamiltonians can be understood in terms of the Berry curvature 2-form, which gives a non-zero quantized Chern number when integrated over $S^2$. Another approach focuses on the family of quantum ground states over $S^2$; mathematically, this defines a line bundle over $S^2$ (we review some basics of line bundles below). Line bundles over $X$ are classified by their first Chern class, an element of the cohomology group $H^2(X, \mathbb{Z})$, where here $X = S^2$. Physically, a non-vanishing first Chern class is an obstruction to making a continuous global choice of the phase of the ground state wave function over $S^2$. Moreover, the Chern number as obtained from Berry curvature can also be viewed as an element of $H^2(S^2, \mathbb{Z})$,[3] so we have two ways to obtain the first Chern class, each using distinct geometric objects, the Berry curvature on one hand, and the ground state line bundle on the other, as illustrated in Figure 1(a).

There are many higher-dimensional analogues of the spin-1/2 particle in a magnetic field. The most familiar of these is the Thouless charge pump [3] in $d = 1$, where a quantized amount of conserved U(1) charge is pumped across the system upon adiabatically tuning a parameter $\theta \in S^1$ through a cycle. The Thouless pump is a system over $S^1$, and can be thought of as a means to probe the topology of the infinite-dimensional spaces $\mathfrak{GH}$ or $\mathcal{Q}$ for gapped systems in $d = 1$ with a U(1) symmetry imposed. In particular, the non-triviality of the Thouless charge pump implies that the appropriate connected component of these spaces has non-trivial $\pi_1$.

Recent years have led to new examples beyond the Thouless charge pump, and a deeper understanding thereof [4–18]. In particular, Kapustin and Spodyneiko generalized the Berry curvature 2-form to a higher Berry curvature $(d + 2)$-form for gapped systems in $d$ spatial dimensions [4]. For a family of Hamiltonians whose parameters lie in a $(d + 2)$-dimensional space $X$, integrating the higher Berry curvature over $X$ gives an invariant that is believed to be quantized and take values in (the free part of) $H^{d+2}(X, \mathbb{Z})$, generalizing the Chern number to gapped systems in $d \geq 1$.[4]

---

[3]Strictly speaking, the integrated Berry curvature gives the free part of the Chern class. However, the data of both the free and torsion parts of the Chern class are contained in the Berry connection.

[4]The Higher Berry curvature invariant of gapped families over $X$ can only be identified with $H^{d+2}(X, \mathbb{Z})$ for low

These developments leave open the following questions that we address in this paper:

1. In dimension $d = 0$, the ground state line bundle characterizes the invertible phase over $X$. In $d \geq 1$, what geometric object plays an analogous role?

2. What feature of the description of the ground state over $X$ is obstructed when the $H^{d+2}(X, \mathbb{Z})$ class is non-trivial?

One might guess that in $d \geq 1$ we should again construct a line bundle of ground states, but this is not the right approach, making the problem more interesting. Perhaps most simply, the first Chern class valued in $H^2(X, \mathbb{Z})$ is the only characteristic class of a line bundle, but we want a class in $H^{d+2}(X, \mathbb{Z})$. Moreover, the family of ground states of a spatially infinite system does not even naturally assemble into a line bundle, because different ground states do not lie in the same Hilbert space.[5] Therefore, a different kind of mathematical object is needed.

Here we consider parametrized families of ground states in $d = 1$ that can be described as matrix product states (MPS). Related work studying parametrized systems using MPS can be found in Refs. [14, 16, 17]. While we assume that each ground state can be written as an injective MPS, it turns out to be essential not to assume the bond dimension of the injective MPS tensor is constant as parameters are varied. Given such a family parametrized over a space $X$, we construct a mathematical object known as a gerbe over $X$. A gerbe is a generalization of a line bundle associated with a class valued in $H^3(X, \mathbb{Z})$ known as the Dixmier-Douady class. In this context, a non-trivial Dixmier-Douady class is an obstruction to finding an MPS tensor which is defined continuously over all of $X$. See Figure 1(b) for an illustration.

In addition, we consider higher dimensional generalizations, where MPS are replaced with projected entangled pair states (PEPS). In $d = 2$ we argue that a parametrized family of PEPS results in a 2-gerbe, a higher generalization of a gerbe, and give an explicit example. This construction is based on the gerbe associated to a parametrized family of MPS, and suggests an inductive construction of $d$-gerbes in $d$-dimensional parametrized families of PEPS.

## 1.2 Background: Parametrized systems

In the above discussion, we considered various parametrized families of Hamiltonians or ground states. Such families are *parametrized quantum systems*, which we now define more formally following Ref. [20]. See also Ref. [21] for a mathematically rigorous treatment. We fix a spatial dimension $d$, a symmetry group $G$, and a choice of whether to consider bosonic or fermionic systems. In this paper, we focus on bosonic systems with trivial symmetry $G$. Leaving out some technical details, these choices specify a space of gapped local Hamiltonians $\mathfrak{GH}$ and a corresponding space of ground states $\mathcal{Q}$.[6] A parametrized quantum system, or a *system over $X$*, is then a continuous map

$$\omega : X \to \mathcal{Q} \,, \tag{2}$$

where $\omega(x)$ is the ground state for $x \in X$, and where $X$ is a topological space that we think of as a space of tuneable parameters. Typically $X$ is some nice, finite-dimensional space, and

---

dimensions. The map $H^{d+2}(X, \mathbb{Z}) \to E^{d+1}(X)$ to the true classification is neither injective nor surjective in general. This is similar to the breakdown of the cohomology classification of SPT phases in higher dimensions.

[5]Condensed matter physics readers might find the statement that different ground states do not lie in the same Hilbert space perplexing; it is an under-appreciated mathematical fact. In condensed matter physics we often work with finite systems and take the limit of a large system; this is of course a very useful viewpoint but it does not provide a mathematical definition of the Hilbert space in the limit. One might think it is possible to define the infinite-system Hilbert space as an infinite tensor product, but this turns out not to be mathematically sensible. Instead, it is possible to use the technology of $C^*$-algebras to work directly with infinite systems, in which case the Hilbert space is constructed *from* the ground state via the Gelfand-Naimark-Segal construction. See Ref. [19] for an accessible treatment.

[6]Depending on the precise setup desired, supplying some of the technical details is an open problem. In particular, we are not aware of a mathematically rigorous construction of the space $\mathfrak{GH}$.

is referred to as the parameter space; however, $X$ should be distinguished from the infinite-dimensional parameter space $\mathfrak{G}\mathfrak{H}$. For the spin-1/2 in a Zeeman field, $X = S^2$, while for the Thouless charge pump, $X = S^1$. Alternatively, and essentially equivalently, we can define a system over $X$ as a continuous map $H : X \to \mathfrak{G}\mathfrak{H}$. In this paper we use these two definitions interchangeably but emphasize parametrized systems defined as families of ground states.

Parametrized quantum systems are of interest in part as a means to probe the topology of the infinite-dimensional spaces $\mathcal{Q}$ and $\mathfrak{G}\mathfrak{H}$. This is closely related to the notion of parametrized phases. Two systems $\omega_0(x)$ and $\omega_1(x)$ over $X$ are said to be in the same phase over $X$ if the maps $\omega_0 : X \to \mathcal{Q}$ and $\omega_1 : X \to \mathcal{Q}$ are homotopic. That is, there exists a continuous function $\omega(x, t) \in \mathcal{Q}$ where $x \in X$ and $t \in [0, 1]$, and such that $\omega(x, 0) = \omega_0(x)$ and $\omega(x, 1) = \omega_1(x)$. Parametrized phases over $X$ are thus nothing but homotopy classes of maps $[X, \mathcal{Q}]$; taking $X = S^n$, these homotopy classes are simply elements of the homotopy groups $\pi_n$. For simplicity, we have ignored the important role of stabilization by stacking with trivial systems (defined below). We refer the reader to Refs. [20] and [21] for discussions from different points of view, and for more details on parametrized phases. The latter reference describes a setup where stacking stabilization is built into a construction of the space $\mathcal{Q}$, and in this setup parametrized phases really are homotopy classes of maps.

We define the trivial system over $X$ to be one where the map $\omega$ is constant as a function of $x \in X$ and $\omega(x)$ is a product state over individual lattice sites. A system over $X$ is said to be nontrivial if it does not belong to the same phase as the trivial system over $X$.

Beyond probing the topology of $\mathcal{Q}$ and $\mathfrak{G}\mathfrak{H}$, parametrized systems are physically interesting in their own right. We can think of a parametrized system as modeling a system with one or more continuously tuneable parameters, and parametrized phases as capturing associated universal phenomena. For example, in Ref. [20], some of the authors introduced a solvable $d = 1$ spin system over $S^3$, and analyzed the quantized pumping of Berry curvature across the system. This system will play an important role in this paper, and following Ref. [20] we refer to it as a Chern number pump.[7] With open boundary conditions, each end of the $d = 1$ chain has a single gapless Weyl point as parameters are varied over $S^3$. Such behavior is not possible for a system in $d = 0$, and reflects an "anomaly in the space of coupling constants" as in Ref. [22].

Finally, parametrized systems are also of interest in connection with Kitaev's proposal that spaces of $d$-dimensional gapped invertible systems form a loop spectrum [23, 25, 26]. This implies that (non-parametrized) invertible phases in $d$ dimensions are classified by a generalized cohomology theory $E^d(\text{pt})$, where pt is the single-point topological space. If instead we evaluate the same cohomology theory on some more interesting space $X$, then $E^d(X)$ gives the classification of invertible phases over $X$.

## 1.3 Summary of results

We now give a high-level overview of the constructions employed in this paper and their implications. It is instructive to present our results in analogy with the more familiar case of $d = 0$ systems, and we begin by reviewing this case. In particular, we describe the construction of the ground state line bundle over $X$, which in turn completely characterizes the phase over $X$.

Let $\mathcal{H}$ be the $d = 0$ system's finite-dimensional Hilbert space, and let $\{U_a\}$ be a cover of $X$ by open sets. We suppose that for each open set there is a continuous function $\Psi_a : U_a \to \mathcal{H}$ whose value $\Psi_a(x)$ is the ground state wave function of the Hamiltonian $H(x)$. Now consider a double overlap $U_{ab} = U_a \cap U_b$. For $x \in U_{ab}$, we must have $\Psi_a(x) = g_{ab}(x)\Psi_b(x)$, where $g_{ab}(x)$ is a non-zero complex number. This statement holds because physical states are rays in

---

[7]Ref. [20] more strictly reserved the term *Chern number pump* for a closely related system over $S^2 \times S^1$, but the essential physics of both systems is the same, so here we use the term pumping somewhat more loosely.

Hilbert space, so while $\Psi_a(x)$ and $\Psi_b(x)$ need not be the same vector in Hilbert space, they both represent the ground state of $H(x)$ and must lie in the same ray. Therefore, for every double overlap, we have a continuous map $g_{ab} : U_{ab} \to \mathbb{C}^\times$, where $\mathbb{C}^\times$ is the multiplicative group of non-zero complex numbers. This is illustrated in the left panel of Fig. 2. The transition functions satisfy a compatibility condition on triple overlaps $U_{abc} = U_a \cap U_b \cap U_c$, namely $g_{ac}(x) = g_{ab}(x)g_{bc}(x)$ for all $x \in U_{abc}$.

A line bundle over $X$ can be specified by giving an open cover $\{U_a\}$ and a set of transition functions satisfying the compatibility condition. Essentially equivalently, the same data specifies a principal $\mathbb{C}^\times$-bundle over $X$; we will mainly work with principal $\mathbb{C}^\times$-bundles in this paper. Note that, above, the local ground state wave functions $\Psi_a$ are only used as an intermediate step to obtain the $g_{ab}$ transition functions; the $\Psi_a$ are not actually needed as part of the data used to specify the ground state line bundle.

For concreteness, we examine the spin-1/2 particle in a Zeeman field (1). Cover $X = S^2$ with two open sets $U_N = S^2 \setminus (0, 0, -1)$ and $U_S = S^2 \setminus (0, 0, 1)$. The overlap $U_N \cap U_S$ is homotopy equivalent to the equatorial $S^1$. On the two open sets, the ground state wave function can be written as

$$
\begin{aligned}
|\psi_N(\theta, \phi)\rangle &= -\sin\frac{\theta}{2} e^{-i\phi}|\uparrow\rangle + \cos\frac{\theta}{2}|\downarrow\rangle, \\
|\psi_S(\theta, \phi)\rangle &= -\sin\frac{\theta}{2}|\uparrow\rangle + \cos\frac{\theta}{2} e^{i\phi}|\downarrow\rangle.
\end{aligned}
\tag{3}
$$

Note that neither wave function can be extended continuously over all of $S^2$. For instance, $|\psi_N(\theta, \phi)\rangle$ depends on $\phi$ as the south pole $\theta = \pi$ is approached, so there is no way to extend $|\psi_N(\theta, \phi)\rangle$ to a function continuous at the south pole. On the overlap $U_N \cap U_S$, the two wave

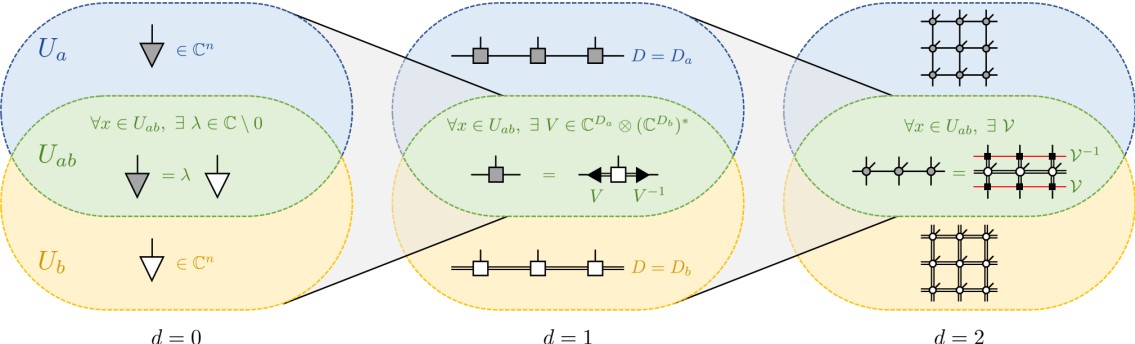

Figure 2: Schematic summary of the constructions of line bundles, gerbes, and 2-gerbes in terms of tensor networks in spatial dimensions $d = 0, 1, 2$, respectively. Within each open set $U_a$, we continuously choose a few-body wavefunction (a $d = 0$ tensor network), an MPS, or a PEPS, representing the states of a parametrized family. In $d$-dimensions, within double overlaps $U_{ab} = U_a \cap U_b$, the object which relates the representations from the two open sets is essentially the same object that was defined on open sets in $(d-1)$-dimensions, leading to a hierarchical structure. For example, in $d = 1$ we have a MPS tensor of bond dimension $D_a$ in each open set $U_a$, from which we obtain a matrix $V \in \mathbb{C}^{D_a} \otimes (\mathbb{C}^{D_b})^*$ on double overlaps relating the two MPS. (More precisely, this holds when one of the MPS is injective on $U_{ab}$.) This matrix can be thought of as a vector in $\mathbb{C}^{D_a D_b}$, the same kind of object that appears within each open set for the $d = 0$ family, thus relating the middle and left panels. The right and middle panels are similarly related. In $d = 2$, we have a PEPS tensor on each open set $U_a$, and a matrix product operator (MPO) $\mathcal{V}$ relating the two PEPS on each double overlap. The MPO can be viewed in the standard fashion as an MPS tensor, as appears within the open sets for $d = 1$.

functions differ by the transition function

$$g : U_N \cap U_S \to U(1) \subset \mathbb{C}^\times \,,$$
$$(\theta, \phi) \mapsto e^{i\phi} \,,$$

(4)

which has a nonzero winding number around the equatorial $S^1$. The winding number of the transition function encodes the non-triviality of the ground state line bundle over $S^2$, and indicates a fundamental obstruction to making a continuous choice of phase for the ground state over all of $S^2$. Here there is only a single double overlap and no triple overlaps, so the compatibility condition on transition functions is superfluous.

In this paper, we describe how nontrivial families of one-dimensional states can be understood in a "higher" analogue of the above picture, illustrated schematically in Fig. 2. To do so we make use of the formalism of matrix product states (MPS) [27], which also play a fundamental role in the classification of one-dimensional phases of matter [28–30]. An MPS provides a description of a $d = 1$ ground state in terms of a three-index tensor $A$, as we describe in more detail in Sec. 2. We show how the formalism of MPS allows us to naturally uncover the structure of a *gerbe*, which is the appropriate generalization of a line bundle to $d = 1$ parameterized systems.

To derive the gerbe structure, we begin with an open cover $\{U_a\}$ of $X$. We assume that on each $U_a$ there exists a continuous MPS tensor $A(x)$, which represents the ground state as a function of parameters locally on $X$.[8]. On double overlaps $U_{ab}$, two MPS tensors are related not by a transition function, but by a transition *line bundle* (or principal $\mathbb{C}^\times$-bundle). This bundle arises from the gauge degree of freedom in representing a ground state with an MPS tensor: for a given ground state, the choice of MPS tensor is not unique, as one can perform matrix-valued gauge transformations on the tensor without changing the state it describes. Under certain assumptions, this gauge transformation is unique up to a choice of a non-zero complex number. Even though this number need not have unit modulus, *i.e.* it need not be an element of U(1), we refer to it as a phase. Therefore, at every point in $U_{ab}$, we have two MPS tensors that are related by a gauge transformation which is unique up to a phase choice. This structure is drawn in the middle panel of Fig. 2. The resulting phase degree of freedom at each point in $U_{ab}$ suggests the structure of a line bundle. In order to capture all types of non-trivial $d = 1$ parametrized phases, we will find it is necessary to go beyond the familiar case where gauge transformations are unique up to $\mathbb{C}^\times$, which introduces several technical challenges, but the same intuition holds nonetheless.

The transition line bundles on double overlaps encode, in part, the topological information of a gerbe, together with some additional data (certain bundle isomorphisms on triple overlaps) and a compatibility condition on quadruple overlaps, as reviewed in Sec. 5.1. A gerbe is characterized by an element of $H^3(X, \mathbb{Z})$ known as its Dixmier-Douady class, and a gerbe is trivial (by definition) when this class vanishes. If it is possible to choose a continuous MPS tensor $A(x)$ globally over $X$, then we obtain a trivial gerbe. Therefore, a non-trivial gerbe indicates an obstruction to a continuous global choice of MPS tensor over $X$. In other words, for non-trivial $d = 1$ systems over $X$, even when the ground state can be exactly described as an MPS tensor $A(x)$ for all $x \in X$, it is nevertheless impossible to choose $A(x)$ continuously everywhere.

An important feature of our construction is that it captures both torsion and non-torsion (*i.e.* free) classes in $H^3(X, \mathbb{Z})$. This is different from other constructions that have appeared. The structure of the space of MPS has been previously studied in Ref. [32], where the authors described a $PGL(\chi)$-bundle structure under the assumption that the injective MPS bond dimension (*i.e.* the bond dimension when the tensor is injective) is constant with value $\chi$ over

---

[8]What we call a "continuous MPS tensor" is not to be confused with cMPS, a generalization of MPS used to describe field theories [31]

$X$. Our construction shows that such a bundle structure restricts the types of classes that can be realized to be torsion only, and that this restriction is removed by allowing the injective bond dimension to vary over $X$.

As a key example that captures a non-torsion class, we illustrate the construction of a gerbe for the $d = 1$ Chern number pump over $X = S^3$ introduced in Ref. [20]. We cover $S^3$ with two open sets $U_N$ and $U_S$ whose overlap $U_N \cap U_S$ includes and is homotopy equivalent to the equatorial $S^2$, and we define continuous MPS tensors on each open set. On the overlap, the transition line bundle is given by the non-trivial line bundle describing the ground state of the spin-$1/2$ particle in a Zeeman field as described above. In this case there are no triple or quadruple overlaps, and the non-triviality of the line bundle signals the non-triviality of the gerbe. Crucially, the injective bond dimension of the tensors differs between the two open sets, such that there is no $PGL(\chi)$-bundle structure, and the corresponding Dixmier-Douady class is non-torsion.

As an example that captures a torsion class, we also construct a gerbe associated to a nontrivial $d = 1$ system over $X = \mathbb{R}P^2 \times S^1$, which we introduce. The system over $X$ is constructed using the suspension construction of Ref. [20] and can be interpreted as a pump of the nontrivial $d = 0$ system over $\mathbb{R}P^2$. Like in the example over $S^3$, we can cover $\mathbb{R}P^2 \times S^1$ with open sets $U_0$ and $U_1$ and define continuous MPS tensors on each set. The overlap $U_0 \cap U_1$ is homotopy equivalent to $\mathbb{R}P^2 \times S^0$, and the transition line bundle relating the MPS tensors on $U_0$ and $U_1$ is given by the ground state line bundle of a nontrivial $0d$ system over $\mathbb{R}P^2$. In this case, the injective bond dimension is constant across $X$ with $\chi = 2$, so there is a $PGL(2)$-bundle structure. The Dixmier-Douady class is torsion, but it is non-zero and this signifies an obstruction to lifting the $PGL(2)$-bundle to a $GL(2)$-bundle.

The ideas sketched above apply to higher dimensional systems as well. The higher-dimensional analogue of MPS is given by projected entangled pair states (PEPS) which, in two-dimensions, represent the ground state in terms of a five-index tensor [27]. Similar to MPS, different PEPS tensors can describe the same state. In this case, under certain technical and physical assumptions, the object relating two PEPS tensors describing the same state can be roughly viewed as a $d = 1$ state that can be exactly described as an MPS. Therefore, if we repeat the above construction by defining continuous PEPS tensors on an open cover of $X$, then on each double overlap $U_{ab}$ we effectively have a family of $d = 1$ states representable as MPS, which we just argued defines a gerbe. This is illustrated in the right panel of Fig. 2. The geometrical object built from gerbes on double overlaps is called a 2-gerbe, and equivalence classes of 2-gerbes are indeed described by elements of $H^4(X, \mathbb{Z})$ [33], which matches the "within-cohomology" part of the conjectured classification of $d = 2$ invertible parameterized systems. Since PEPS representations are much less well understood than for MPS, we do not give a rigorous derivation of this 2-gerbe in general, but we are able to show that the non-trivial $d = 2$ parameterized system over $S^4$ given in Ref. [20] indeed realizes a non-trivial 2-gerbe. Finally, we discuss how this perspective suggests a natural construction of $d$-gerbes associated to parameterized systems of $d$-dimensional PEPS.

The structure of the paper is as follows. In Sec. 2 we provide a primer on the formalism of matrix product states. Sec. 3 reviews the expectation for the classification of parametrized systems from the perspective of homotopy theory, and explains why we must go beyond injective MPS of fixed bond dimension to recover the full classification. We analyze the Chern number pump in Sec. 4 and provide an MPS representation for the family of states. The general procedure for constructing a gerbe given an MPS family is then given in Sec. 5. We then analyze our construction in the special case of families where the injective bond dimension $\chi$ is constant over $X$, and show that it gives rise to a $PGL(\chi)$-bundle as expected. We provide a non-trivial such example – a system over $\mathbb{R}P^2 \times S^1$ – in Sec. 6. Finally, in Sec. 7, we sketch how the construction extends to higher dimensions using PEPS, and illustrate the ideas using

a $d = 2$ parametrized system over $S^4$ which also appeared in Ref. [20]. We conclude with a discussion in Sec. 8.

## 2  Matrix product state generalities

In this paper we will consider families of states which admit translationally invariant MPS representations. While in this paper we are mainly interested in infinite systems, it is convenient for a moment to work on a periodic one dimensional lattice with $N$ sites, where each on-site Hilbert space is $\mathbb{C}^n$. A state admits a translationally invariant MPS representation if it is of the form

$$|\psi_N(A)\rangle = \sum_{i_1,\dots,i_N} \mathrm{Tr}\left(A^{i_1} A^{i_2} \dots A^{i_N}\right)|i_1, i_2, \dots, i_N\rangle, \tag{5}$$

for some MPS tensor $A^i_{\alpha\beta}$, $i \in \{0, \dots, n-1\}$, and $\alpha, \beta \in \{1, \dots, D\}$. For each $i$, $A^i$ is a $D \times D$ dimensional matrix acting on the so-called virtual space $\mathbb{C}^D$; the constant $D$ is known as the bond dimension. If the tensor $A^i$ is injective (see below), it defines a pure state of an infinite system that we denote by $\omega_A$ [34, 35]. Somewhat heuristically, the state $\omega_A$ can be thought of as the $N \to \infty$ limit of the states $|\psi_N(A)\rangle$.[9] We say that the injective MPS tensor $A^i$ represents or specifies the state $\omega_A$.

It will be useful throughout this paper to use the graphical notation of tensor networks. To this end, we represent the tensor $A$ in the following way,

$$A^i_{\alpha\beta} = \quad \alpha \overset{\overset{i}{|}}{\underset{A}{\bullet}} \beta \quad . \tag{6}$$

The coefficients of the wavefunction in Eq. (5) are then expressed graphically as

$$\mathrm{Tr}\left(A^{i_1} A^{i_2} \dots A^{i_N}\right) = \quad \overset{i_1}{\underset{A}{\bullet}} \overset{i_2}{\underset{A}{\bullet}} \dots \overset{i_N}{\underset{A}{\bullet}} \quad , \tag{7}$$

where contracted indices are summed over, and the curved lines at the end connect to each other, representing periodic boundaries.

An MPS tensor $A^i$ can be viewed as a linear map $A : M_D(\mathbb{C}) \to \mathbb{C}^n$ from linear operators on the virtual space $\mathbb{C}^D$ to the physical space $\mathbb{C}^n$, given by $M \mapsto \sum_i \mathrm{Tr}(A^i M^T)|i\rangle$. The MPS tensor $A^i$ is said to be *injective* if this associated map is injective [27]. Injective MPS tensors satisfy many nice properties, such as having a finite correlation length and being the unique ground state of their (canonical) parent Hamiltonian. A more general class of tensors is given by *normal tensors*, for which the map $M \mapsto \sum_{i_1,\dots,i_L} \mathrm{Tr}(A^{i_1} \dots A^{i_L} M^T)|i_1, \dots, i_L\rangle$ becomes injective for some $L$. Given a normal tensor, the minimum such $L$ is known as the *injectivity length* and is bounded above by a function that depends only on the bond dimension. [36] It follows that normal tensors become injective upon "blocking" sites together into new sites comprised of $L$ original sites; that is, the tensor $\tilde{A}^{i_1 \dots i_L} \equiv A^{i_1} \cdots A^{i_L}$ is injective. Because this can be done, in this paper we generally do not work with tensors that are normal but not injective.[10]

For a state $\omega$ with a translationally invariant MPS representation, the tensor $A^i$ used to represent the state is not unique. It is clear from (5) that if the injective tensor $A^i$ represents

---

[9]Formally, a state $\omega$ of an infinite system is a positive linear functional on the operator algebra of observables. If $O$ is a local operator, then $\omega(O)$ is interpreted as the expectation value of $O$. The state $\omega_A$ obtained from an MPS tensor belongs to a family of states known as finitely correlated states [34].

[10]Given our emphasis on injective tensors, we state results from the MPS literature in terms of injective tensors even when the same result holds for the weaker condition of normality.

$\omega$, then so does $\lambda M A^i M^{-1}$ for any invertible matrix $M \in GL(D) \equiv GL(D, \mathbb{C})$ and any nonzero complex number $\lambda \in \mathbb{C}^\times$, i.e. $\omega_A = \omega_{\lambda M A M^{-1}}$. In fact, the fundamental theorem of injective MPS states that a pair of injective tensors $A$ and $B$ represent the same state if and only if $B^i = \lambda M A^i M^{-1}$ for some $M$ and $\lambda$ [27]. The transformation $A^i \mapsto \lambda M A^i M^{-1}$ is called a gauge transformation of the tensor $A$. In particular, this means $A^i$ and $B^i$ have the same bond dimension, so given a state $\omega$ with an injective MPS representation, the bond dimension of an injective MPS tensor representing $\omega$ is a well-defined quantity – we refer to this as the *injective bond dimension of* $\omega$ and denote it by $\chi_\omega$. Moreover, given $A^i$ and $B^i$, $\lambda$ is uniquely specified, and $M$ is unique up to a nonzero scalar multiple, $M \mapsto zM$ with $z \in \mathbb{C}^\times$. This means that, while $M \in GL(D)$ is not uniquely specified given $A^i$ and $B^i$, the corresponding element of the projective general linear group $PGL(D) = PGL(D, \mathbb{C}) = GL(D)/\mathbb{C}^\times$ is uniquely specified. To summarize, $A^i$ and $B^i$ are related by a unique element of $\mathbb{C}^\times \times PGL(D)$.

In this paper we will also need to discuss MPS tensors which are not injective.[11] Given a state $\omega$ represented by an injective tensor $A^i$, and a not necessarily injective tensor $B^i$, we say that $B^i$ represents the state $\omega$ if

$$|\psi_N(B)\rangle = \lambda^N |\psi_N(A)\rangle \,, \tag{8}$$

for some $\lambda \in \mathbb{C}^\times$ for all $N$. While one can imagine weaker definitions of what it means for $B^i$ to represent $\omega$, working with this class of tensors allows us to take advantage of an important technical tool known as a *reduction* [37]. In Ref. [37], it is shown that if Eq. (8) holds, then there exist matrices $V$ and $W$ such that $VW = \mathbf{1}$ and

$$V B^{i_1} \dots B^{i_k} W = \lambda^n A^{i_1} \dots A^{i_k} \,, \tag{9}$$

for the same $\lambda$ appearing in (8), and any string $i_1 \dots i_k$. The pair $V, W$ is called a reduction from $B^i$ to $A^i$. We discuss this further in Sections 4 and 5.

We conclude this section with a brief introduction to some diagrammatic notations which will facilitate many of our computations. For the purposes of this discussion we will restrict to the case where the on-site Hilbert space is $\mathbb{C}^2$, though of course this is easily generalized. The key concept is the correspondence between operators and states. A single-qubit operator can be turned into an entangled state by "bending" the legs of the tensor:

$$O \,\bullet \quad \Longrightarrow \quad O \,\bullet \quad \Big| \quad . \tag{10}$$

The meaning of this entangled state becomes clear when considering the special case where $O$ is the identity. In this case we have the correspondence

$$\Big| \quad \Longrightarrow \quad \Big|\_\Big| \quad = |\Phi^+\rangle \,, \tag{11}$$

which relates the identity operator with the Bell state $|\Phi^+\rangle = |\uparrow\uparrow\rangle + |\downarrow\downarrow\rangle$. Therefore the state corresponding to the single-qubit operator $O$ in (10) is simply $(O \otimes \mathbf{1})|\Phi^+\rangle$. The state $|\Phi^+\rangle$ also satisfies the useful property $(O \otimes \mathbf{1})|\Phi^+\rangle = (\mathbf{1} \otimes O^T)|\Phi^+\rangle$, which allows us to push an operator through a Bell state,

$$O \,\bullet \quad \Big| \quad = \quad \Big| \quad \bullet\, O^T \quad . \tag{12}$$

For notational convenience, we will also sometimes write the operator acting on the Bell state in the middle of the tensor. We define this to mean that the operator acts on the first qubit of the Bell state:

$$\Big|\_{\bullet}\Big| \quad = O \,\bullet \quad \Big| \quad . \tag{13}$$

$$O$$

---

[11]This does not mean we are considering states with symmetry breaking or long-range correlations, which are necessarily non-injective. Instead, we will need to consider MPS representations that are not in canonical form.

# 3 Topology of SRE states and MPS

Short range entangled phases in one spatial dimension are expected to be classified by the Eilenberg-MacLane space $K(\mathbb{Z}, 3)$. This means that phases parametrized by $X$ should be in correspondence with homotopy classes of maps $[X, K(\mathbb{Z}, 3)] \cong H^3(X, \mathbb{Z})$. Ref. [4] introduced a curvature three-form which, when integrated over the parameter space $X$, detects the free part of the class in $H^3(X, \mathbb{Z})$.

Let $\mathcal{M}_\chi$ be the space of states of an infinite system which are representable by injective MPS of bond dimension $\chi$. Physically, such states have finite correlation length and are short-range entangled. For finite systems, it was proven in Ref. [32] that the set of injective MPS tensors $\mathcal{A}_\chi$ of bond dimension $\chi$ forms a principal $\mathbb{C}^\times \times PGL(\chi)$-bundle over $\mathcal{M}_\chi$. We expect the same result to hold for infinite systems but to our knowledge it has not been proved rigorously. The intuition for this is given by the fundamental theorem of MPS (see Sec. 2): any two MPS tensors representing the same state are related by a unique gauge transformation living in the group $\mathbb{C}^\times \times PGL(\chi)$, so the action is free and transitive on the fibers. In Appendix B, we prove that, if we allow the onsite dimension to go to infinity, $\mathcal{M}_\chi$ is in fact equivalent to the classifying space $B(PGL(\chi) \times \mathbb{C}^\times)$.

A family of states over $X$ representable by injective MPS of bond dimension $\chi$ can be thought of as a map

$$X \to \mathcal{M}_\chi. \tag{14}$$

Using this map we can pull back the canonical principal $\mathbb{C}^\times \times PGL(\chi)$-bundle over $\mathcal{M}_\chi$ to obtain a principal $\mathbb{C}^\times \times PGL(\chi)$-bundle over $X$. This is equivalent to two separate $\mathbb{C}^\times$ and $PGL(\chi)$ principal bundles over $X$, and we discuss these in turn.

Principal $\mathbb{C}^\times$-bundles are classified by the first Chern class $H^2(X, \mathbb{Z})$. To identify a physical interpretation of this Chern class, we consider the $d = 1$ system over $X = S^2$ obtained by placing a decoupled spin-1/2 particle in a Zeeman field (Equation (1)) on each lattice site. Representing this system as an injective MPS with bond dimension $\chi = 1$ and constructing the corresponding $\mathbb{C}^\times$-bundle, one obtains a class that generates $H^2(S^2, \mathbb{Z}) \cong \mathbb{Z}$, and which is simply the Chern class of a single spin-1/2 particle. We thus identify the $H^2(X, \mathbb{Z})$ class as a "Chern number per crystalline unit cell," which is expected to be a well-defined phase invariant for a system with translation symmetry. We will for the most part not be interested in this $H^2(X, \mathbb{Z})$ invariant. Indeed, to obtain a nice definition of $d = 1$ parametrized phases without translation symmetry, we expect it is necessary to take a quotient by suitable decoupled systems to eliminate the $H^2(X, \mathbb{Z})$ invariant.

We now turn to the $PGL(\chi)$-bundle. It is natural to ask whether nontrivial such bundles correspond to nontrivial parametrized systems (ignoring the $H^2(X, \mathbb{Z})$ invariant). We argue that $PGL(\chi)$-bundles cannot capture all parametrized phases; in particular, they do not capture the known nontrivial phases over $S^3$ [4, 20], expected to be classified by $[S^3, K(\mathbb{Z}, 3)] \cong H^3(S^3, \mathbb{Z}) \cong \mathbb{Z}$. Principal $G$-bundles over $S^3$ are classified, via the clutching construction, by homotopy classes of maps $[S^2, G]$. But $\pi_2(PGL(\chi)) = 0$, so no nontrivial bundle exists.

This result strongly suggests that general nontrivial $d = 1$ parametrized phases cannot be described using injective MPS with a fixed bond dimension. One option is to allow for families of states where the injective bond dimension varies as a function of parameters. Indeed, we will see in Sec. 4 that this is precisely what happens in our main example of the Chern number pump over $S^3$, where the injective bond dimension takes values $\chi = 1, 2$ depending on $x \in S^3$. This system can thus be thought of as a map $S^3 \to \mathcal{M}_1 \cup \mathcal{M}_2$. It is an important point that this union of $\mathcal{M}_1$ and $\mathcal{M}_2$ should not be disjoint; this follows from the construction of Sec. 4.[12]

---

[12]Formally, $\mathcal{M}_\chi$, and unions thereof for different values of $\chi$, should be viewed as a subspace of the space of all pure states given the weak-* topology.

Because $\chi$ varies, there is no longer an obvious principal bundle structure over $\mathcal{M}_1 \cup \mathcal{M}_2$, which in part leads us to our construction of a gerbe in Sec. 5.

Nevertheless, it is possible for some $X$ to host nontrivial $PGL(\chi)$-bundles. Moreover, to such a principal $PGL(\chi)$-bundle over $X$, one can also assign a gerbe over $X$ called the lifting gerbe; the lifting gerbe measures the obstruction to lifting the $PGL(\chi)$-bundle to a $GL(\chi)$-bundle. It is known [38–40] that these lifting gerbes correspond to torsion elements of $H^3(X, \mathbb{Z})$. We give more details on the relationship between $PGL(\chi)$-bundles and torsion in $H^3(X, \mathbb{Z})$ in Appendix C. This suggests that parametrized systems over spaces $X$ with the property that $H^3(X, \mathbb{Z})$ contains torsion are related to families of injective MPS of constant bond dimension $\chi$, whose associated $PGL(\chi)$-bundle are nontrivial, but do not lift to $GL(\chi)$-bundles. Conversely, parametrized systems whose associated class in $H^3(X, \mathbb{Z})$ is non-torsion necessarily cannot be represented by injective MPS of constant bond dimension. In Section 6, starting from a nontrivial family of states obtained from the suspension construction, [20] we construct a family of injective MPS over $X = \mathbb{R}P^2 \times S^1$ of constant bond dimension $\chi = 2$ with a nontrivial $PGL(2)$-bundle.

Finally, let us comment on a possibility that we leave unaddressed in this work. It is possible that the $PGL(\chi)$-bundle over $X$ associated to a family of states is nontrivial, but has a lift to a nontrivial $GL(\chi)$-bundle. In this case the associated $H^3(X, \mathbb{Z})$ class is trivial, and we expect the parametrized phase to be trivial, but the family of MPS is still topologically distinct from the constant family of MPS. We comment on this possibility further in Appendix C, but leave a detailed analysis to future work.

# 4 MPS representation of Chern number pump

In this section we review the model of the Chern number pump introduced in Ref. [20] and construct an MPS representation for the family of ground states over $S^3$. As noted above, the ground states cannot be described globally by a normal MPS tensor of constant bond dimension. However, this turns out to be a feature rather than a bug; we will leverage the changing bond dimension to describe the phase invariant of the system.

The model is a slight modification of the one given in Ref. [20]. We consider a one dimensional lattice with two qubits per site and take the parameter space to be $X = S^3$. Sites are labeled by $i \in \mathbb{Z}$ and Pauli operators for the two qubits on each site are denoted $\sigma_{i,a}^{x,y,z}$ and $\sigma_{i,b}^{x,y,z}$. We parametrize elements $x \in X$ as $x = (\vec{w}, w_4) \in S^3 \subset \mathbb{R}^4$, where $\vec{w}$ is a three-component vector such that $|\vec{w}|^2 + w_4^2 = 1$. The Hamiltonian takes the form

$$H(\vec{w}, w_4) = \sum_{i \in \mathbb{Z}} \left( H_i^B(\vec{w}) + H_i^+(w_4) + H_i^-(w_4) \right). \tag{15}$$

The first term is an on-site field which takes opposite values on the $A$ and $B$ sublattices,

$$H_i^B(\vec{w}) = \vec{w} \cdot \vec{\sigma}_{i,a} - \vec{w} \cdot \vec{\sigma}_{i,b}. \tag{16}$$

The second and third terms are inter-site and intra-site couplings, with

$$\begin{aligned} H_i^+(w_4) &= g^+(w_4)\, \vec{\sigma}_{i,b} \cdot \vec{\sigma}_{i+1,a}, \\ H_i^-(w_4) &= g^-(w_4)\, \vec{\sigma}_{i,a} \cdot \vec{\sigma}_{i,b}. \end{aligned} \tag{17}$$

The functions $g^\pm(w_4)$ are chosen to be

$$g^+(w_4) = \begin{cases} \sqrt{w_4^2 - 1/4}, & w_4 \geq \frac{1}{2}, \\ 0, & w_4 \leq \frac{1}{2}, \end{cases} \tag{18}$$

and

$$g^-(w_4) = \begin{cases} 0, & w_4 \geq -\frac{1}{2}, \\ \sqrt{w_4^2 - 1/4}, & w_4 \leq -\frac{1}{2}. \end{cases} \tag{19}$$

Note that, for any value of $w_4$, at most one of $H^+(w_4)$ or $H^-(w_4)$ is nonzero. As a result, $H(\vec{w}, w_4)$ is always a sum of decoupled two-qubit dimer Hamiltonians, each of which is exactly solvable. It follows that the ground state is a product state of dimers, where the dimerization pattern depends on the value of $w_4$, and consists of inter-site dimers for $w_4 \geq 0$ and intra-site dimers for $w_4 \leq 0$ (see Fig. 3).

The spectrum of the full Hamiltonian is completely determined by the spectrum of the zero-dimensional dimers, so it is easy to see that there is a unique gapped ground state everywhere on $S^3$. We cover $S^3$ by two open sets

$$\begin{aligned} U_N &= \left\{ (\vec{w}, w_4) \in S^3 \,\middle|\, w_4 > -\frac{1}{2} \right\}, \\ U_S &= \left\{ (\vec{w}, w_4) \in S^3 \,\middle|\, w_4 < \frac{1}{2} \right\}. \end{aligned} \tag{20}$$

The overlap $U_{NS} = U_N \cap U_S$ can be viewed as a thickened version of the equatorial $S^2$ defined by $w_4 = 0$. Indeed, $U_{NS}$ is homotopy equivalent to $S^2$. The ground state of each dimer can be written in the following form:

$$\begin{aligned} |\psi\rangle_{dimer}^{N/S} &= \left( U(\vec{w}) \otimes U(\vec{w}) \right) \left( \Lambda^{N/S}(\vec{w}) \otimes \mathbf{1} \right) |\Phi^+\rangle \\ &= \left( U(\vec{w}) \Lambda^{N/S} U(\vec{w})^T \otimes \mathbf{1} \right) |\Phi^+\rangle \\ &= \tilde{\Lambda}^{N/S} \, \vert\!\!\raisebox{0.3ex}{\rule{1.2em}{0.4pt}}\!\!\vert \,. \end{aligned} \tag{21}$$

This state $|\psi\rangle_{dimer}^N$ is defined on $U_N$ and lives on inter-site dimers, while $|\psi\rangle_{dimer}^S$ lives on intra-site dimers and is defined on $U_S$ (see Fig. 3). Here, $|\Phi^+\rangle = \frac{1}{\sqrt{2}}(|\uparrow\uparrow\rangle + |\downarrow\downarrow\rangle)$ is the Bell state as in (11). The $U(\vec{w})$ are single-site rotation matrices

$$U(\theta, \phi) = \begin{pmatrix} \cos\frac{\theta}{2} & -\sin\frac{\theta}{2} e^{-i\phi} \\ \sin\frac{\theta}{2} e^{i\phi} & \cos\frac{\theta}{2} \end{pmatrix}, \tag{22}$$

which rotate eigenstates of $\sigma^z$ to eigenstates of $\hat{w} \cdot \vec{\sigma}$. Here, $\theta$ and $\phi$ are the usual spherical polar coordinates for the unit vector $\hat{w} = \vec{w}/|\vec{w}|$. Note that $U$ is not globally well-defined on

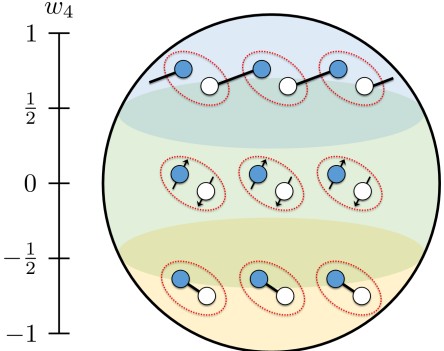

Figure 3: Dimerization pattern for different regions of $w_4$. Unit cells are shown in red, with blue and white sites denoting the $A$ and $B$ sublattices respectively. The dimerization is inter-site for $w_4 \geq 1/2$ and intra-site for $w_4 \leq -1/2$. The middle region $-1/2 \leq w_4 \leq w_4$ is completely factorized.

$S^2$. $\Lambda^N(\vec{w})$ and $\Lambda^S(\vec{w})$ are single-site (non-unitary) operators given by

$$
\Lambda^N(\vec{w}) = \begin{cases} \begin{pmatrix} 0 & -\sqrt{\frac{1}{2} - \frac{|\vec{w}|}{\sqrt{3}}} \\ \sqrt{\frac{1}{2} + \frac{|\vec{w}|}{\sqrt{3}}} & 0 \end{pmatrix}, & w_4 \geq \frac{1}{2}, \\[2em] \begin{pmatrix} 0 & 0 \\ 1 & 0 \end{pmatrix}, & -\frac{1}{2} < w_4 \leq \frac{1}{2}, \end{cases}
\tag{23}
$$

and

$$
\Lambda^S(\vec{w}) = \begin{cases} \begin{pmatrix} 0 & \sqrt{\frac{1}{2} + \frac{|\vec{w}|}{\sqrt{3}}} \\ -\sqrt{\frac{1}{2} - \frac{|\vec{w}|}{\sqrt{3}}} & 0 \end{pmatrix}, & w_4 \leq -\frac{1}{2}, \\[2em] \begin{pmatrix} 0 & 1 \\ 0 & 0 \end{pmatrix}, & -\frac{1}{2} \leq w_4 < \frac{1}{2}. \end{cases}
\tag{24}
$$

Note that $\Lambda^{N/S}(\vec{w})$ are continuous at $|w_4| = 1/2$ since $|\vec{w}| = \sqrt{3}/2$. The operators $\Lambda^{N/S}(\vec{w}) \otimes \mathbf{1}$ send $|\Phi^+\rangle$ to the ground state of the dimer Hamiltonian when $\vec{w}$ is along the $\hat{z}$ axis. Finally we have defined $\tilde{\Lambda}^{N/S}$ to be $U\Lambda^{N/S}U^T$. It can be checked that $\tilde{\Lambda}^{N/S}(\vec{w})$ indeed give well-defined continuous functions on $U_{N/S}$.

The diagrammatic representation of the family of ground states of (15) is formed by tensoring together the diagrammatic representations of the dimers (21). The ground state on $U_N$ is

$$
|GS\rangle = \qquad \tilde{\Lambda}^N \qquad \tilde{\Lambda}^N \qquad \tilde{\Lambda}^N \quad ,
\tag{25}
$$

while the ground state on $U_S$ is

$$
|GS\rangle = \quad \tilde{\Lambda}^S \qquad \tilde{\Lambda}^S \qquad \tilde{\Lambda}^S \quad .
\tag{26}
$$

We have depicted the ground state for finite systems of six qubits with periodic boundary conditions. We can then read off the MPS tensors from the diagram, grouping pairs of qubits into sites as indicated by the dashed boxes. The MPS tensor on $U_N$ is

$$
A_N^{ij} = \qquad \tilde{\Lambda}^N
\tag{27}
$$

$$
= |i\rangle\langle j| U(\vec{w})\Lambda^N(\vec{w})U(\vec{w})^T .
\tag{28}
$$

The MPS tensor on $U_S$ is found similarly as

$$
A_S^{ij} = \quad \tilde{\Lambda}^S
\tag{29}
$$

$$
= \langle i| U(\vec{w})\Lambda^S(\vec{w})U(\vec{w})^T |j\rangle .
\tag{30}
$$

We now make the following observation. The tensor $A_S^{ij}$ has bond dimension $D = 1$ and is injective everywhere it is defined ($w_4 < 1/2$). On the other hand, the tensor $A_N^{ij}$ has bond

dimension $D = 2$ and is defined everywhere on $U_N$ ($w_4 > -1/2$), but is only injective when $w_4 > 1/2$. As a result, on the overlapping region $U_{NS}$, the ground state can be represented both by the injective tensor $A_S^{ij}$ and by the non-injective tensor $A_N^{ij}$. In $U_{NS}$, $\Lambda^N = |\uparrow\rangle\langle\downarrow|$ and $\Lambda^S = |\downarrow\rangle\langle\uparrow|$, so the MPS tensors simplify as

$$
\begin{aligned}
A_N^{ij} &= |i\rangle\langle j|U(\vec{w})|\downarrow\rangle\langle\uparrow|U(\vec{w})^T\,, \\
A_S^{ij} &= \langle i|U(\vec{w})|\uparrow\rangle\langle\downarrow|U(\vec{w})^T|j\rangle\,.
\end{aligned}
\tag{31}
$$

As discussed in Sec. 2, these are related by a reduction from $A_N^{ij}$ to $A_S^{ij}$. The reduction is given by $V = \langle\uparrow|U(\vec{w})^T$, $W = \overline{U(\vec{w})}|\uparrow\rangle$, which satisfies

$$
VA_N^{ij}W = \langle\uparrow|U(\vec{w})^T|i\rangle\langle j|U(\vec{w})|\downarrow\rangle\langle\uparrow|U(\vec{w})^T\overline{U(\vec{w})}|\uparrow\rangle = \langle i|U(\vec{w})|\uparrow\rangle\langle\downarrow|U(\vec{w})^T|j\rangle = A_S^{ij}\,.
\tag{32}
$$

Note, however, that $W$ and $V$ are not well-defined over all of the equatorial $S^2$. The problem occurs at $\theta = \pi$, where the phase is not well-defined. This is essentially the same phase ambiguity that occurs when trying to write down a ground state wavefunction for the spin-1/2 particle in a Zeeman field. We see that there is a nonzero Chern number associated to the reduction from $A_N^{ij}$ to $A_S^{ij}$. This hints at the nontriviality of the family (15); we make this observation more precise in the following section.

# 5 MPS gerbe

In this section we describe the construction of a gerbe from a family of MPS. We begin with a definition of gerbes in Sec. 5.1. Sec. 5.2 is a review of the key technical idea of a reduction, which we extend to the setting of continuous families of MPS. This gives, in some cases, a $\mathbb{C}^\times$-bundle on double overlaps $U_\alpha \cap U_\beta$. This is in fact is sufficient to construct the gerbe of the previous section, which we show in Sec. 5.3. In Sec. 5.4 we proceed to the general construction of the gerbe. Finally, in Sec. 5.5 we apply our construction to the special case where the family of MPS is injective everywhere.

## 5.1 Gerbes

Here we review the aspects of gerbes that are needed in this paper. Gerbes are generalizations of line bundles, or, equivalently, of principal $\mathbb{C}^\times$-bundles. Gerbes were introduced by Giraud in Ref. [41] as an attempt to construct non-abelian cohomology in degree 2, whereas (isomorphism classes of) principal fiber bundles represent non-abelian cohomology in degree 1. In the $\mathbb{C}^\times$-valued case—and this is the only case we consider here—gerbes are classified by degree 3 cohomology with values in $\mathbb{Z}$. There are several equivalent notions of such gerbes, and they all give rise to degree 3 cohomology classes with coefficients in $\mathbb{Z}$, called the Dixmier-Douady classes. We review a formulation based on Ref. [42] which is most naturally suited to an MPS formulation.

In the literature one often sees gerbes defined in terms of line bundles, rather than principal $\mathbb{C}^\times$-bundles as we use below. A natural equivalence between these two concepts is given by assigning to each principal $\mathbb{C}^\times$-bundle $\mathcal{P}$ the associated line bundle $\mathcal{L} = \mathcal{P} \times_{\mathbb{C}^\times} \mathbb{C}$. As pointed out before in Sec. 1.3, these bundles have the same $\mathbb{C}^\times$-valued transition maps. Thus, gerbes may equivalently be defined using line bundles or principal $\mathbb{C}^\times$-bundles. We refer to Ref. [43] for further information on gerbes.

Let $X$ be a compact smooth manifold. A gerbe $\mathcal{P}$ consists of the following data: an open cover $\{U_a\}$, a collection of principal $\mathbb{C}^\times$-bundles $\mathcal{P}_{ab}$ defined on the double intersections

$U_{ab} = U_a \cap U_b$ together with bundle isomorphisms

$$\mathcal{P}_{ab} \cong \mathcal{P}_{ba}^{-1}, \tag{33}$$

where $\mathcal{P}^{-1}$ denotes the inverse of the $\mathbb{C}^\times$-bundle $\mathcal{P}$ (*i.e.* $\mathcal{P}$ with the inverse $\mathbb{C}^\times$-action), and additional bundle isomorphisms

$$\theta_{abc} : \mathcal{P}_{ab} \otimes \mathcal{P}_{bc} \to \mathcal{P}_{ac}, \tag{34}$$

defined on triple intersections $U_{abc} = U_a \cap U_b \cap U_c$. The isomorphisms $\theta_{abc}$ are required to satisfy an associativity condition on quadruple overlaps, namely,

$$\theta_{acd}(\theta_{abc} \otimes 1) = \theta_{abd}(1 \otimes \theta_{bcd}). \tag{35}$$

Together with the isomorphisms of (33), the maps $\theta_{abc}$ give trivializations of the $\mathbb{C}^\times$-bundle $\mathcal{P}_{ab} \otimes \mathcal{P}_{bc} \otimes \mathcal{P}_{ca}$. If $\{U_a\}$ is a good cover, we can pick local sections $s_{ab}$ on each double overlap $U_{ab}$. A choice of local sections defines a function $g_{abc} : U_{abc} \to \mathbb{C}^\times$ over triple intersections via

$$\theta_{abc} \circ (s_{ab} \otimes s_{bc}) = g_{abc} s_{ac}. \tag{36}$$

The associativity condition is equivalent to the condition that

$$g_{bcd} g_{abd} = g_{acd} g_{abc},$$

so that $(g_{abc})$ is a Čech 2-cocycle. Making a different choice of local section $s'_{ab} = f_{ab} s_{ab}$ defines a new function

$$g'_{abc} = f_{ab} f_{bc} f_{ca} g_{abc},$$

so $g_{abc}$ changes by a coboundary. Therefore, it represents a class in $\check{H}^2(X, \mathbb{C}^\times)$. But the exponential sequence

$$\mathbb{Z} \xrightarrow{2\pi i} \mathbb{C} \xrightarrow{\exp} \mathbb{C}^\times,$$

gives rise to a long exact sequence

$$\to \check{H}^k(X, \mathbb{C}) \cong 0 \to \check{H}^k(X, \mathbb{C}^\times) \to H^{k+1}(X, \mathbb{Z}) \to 0 \to,$$

which leads to the isomorphism

$$\check{H}^2(X, \mathbb{C}^\times) \xrightarrow{\cong} H^3(X, \mathbb{Z}).$$

So a gerbe $\mathcal{P} = (\mathcal{P}_{ab})$ defines a cohomology class

$$d(\mathcal{P}) \in H^3(X, \mathbb{Z}).$$

This is the Dixmier-Douady class of the gerbe.

A $\mathbb{C}^\times$-valued Čech 1-cocycle $(g_{ab})$ consists of continuous functions $g_{ab} : U_{ab} \to \mathbb{C}^\times$ which satisfy $g_{ab}^{-1} = g_{ba}$ and the condition $g_{ac} = g_{ab} g_{bc}$ on $U_{abc}$. As discussed in Sec. 1.3, this is nothing but the transition functions for a line bundle. The corresponding class in $\check{H}^1(X, \mathbb{C}^\times) \xrightarrow{\cong} H^2(X, \mathbb{Z})$ is the first Chern class of the line bundle. We thus see how the Dixmier-Douady class is the generalization of the first Chern class of a line bundle.

Like in the $d = 0$ case where the integral of the Berry curvature 2-form recovers the first Chern class of the line bundle of ground states, the Dixmier-Douady class we extract from a family of MPS $d = 1$ is related to the phase invariant obtained by integrating the higher Berry curvature 3-form [4].

## 5.2 MPS reductions

The method we use to construct a gerbe from a family of MPS relies heavily on the notion of a reduction [37], which we review here. We define a $(\chi, D)$-*reduction* to be a pair of matrices $V : \mathbb{C}^D \to \mathbb{C}^\chi, W : \mathbb{C}^\chi \to \mathbb{C}^D$ such that $VW = \mathbf{1}_{\chi \times \chi}$; let $\mathcal{R}_\chi(D)$ denote the space of all $(\chi, D)$-reductions. Note that $VW = \mathbf{1}_{\chi \times \chi}$ implies that $\chi \leq D$. The space $\mathcal{R}_\chi(D)$ can be shown to be homotopy equivalent to the space of $\chi$-frames in $\mathbb{C}^D$, typically denoted by $V_\chi(\mathbb{C}^D)$. This is explained in detail in Appendix A.

It was proven in Ref. [37] that given an MPS tensor $A$ of bond dimension $D$ and an injective MPS tensor $B$ of bond dimension $\chi \leq D$ which represent the same state, there exists a $(\chi, D)$-reduction $V, W$ such that

$$VA^{i_1} \dots A^{i_k}W = \lambda^k B^{i_1} \dots B^{i_k}, \tag{37}$$

for all strings $i_1 \dots i_k$, with $\lambda \in \mathbb{C}^\times$ given by (8). The pair $V$, $W$ satisfying (37) is called a *reduction from A to B*. Let us review the construction of $V$ and $W$. Since $B$ is injective, the corresponding map $B : M_\chi(\mathbb{C}) \to \mathbb{C}^n$ has a left inverse $B^{-1} : \mathbb{C}^n \to M_\chi(\mathbb{C})$:

$$\overset{B^{-1}}{\underset{B}{\rule{0pt}{1pt}}} \;=\; \rceil \; \lceil \;. \tag{38}$$

While this left inverse is not unique, we can take $B^{-1}$ to be the Moore–Penrose inverse,

$$B^{-1} := (B^\dagger B)^{-1} B^\dagger. \tag{39}$$

In the above equation $B$ and $B^\dagger$ are to be understood as maps $B : M_\chi(\mathbb{C}) \to \mathbb{C}^n$ and $B^\dagger : \mathbb{C}^n \to M_\chi(\mathbb{C})$. This gives a canonical choice for the left inverse of $B$.

Next, consider the tensor $\mathcal{T}_{AB} = B^{-1}A$ defined by

$$\mathcal{T}_{AB} = B^{-1}A = \sum_{i=1}^n \left(B^{-1}\right)^i \otimes A^i. \tag{40}$$

Since the MPS tensor components $\left(B^{-1}\right)^i$ and $A^i$ are operators on $\mathbb{C}^\chi$ and $\mathbb{C}^D$, respectively, $\mathcal{T}_{AB}$ can be viewed as an operator on $\mathbb{C}^\chi \otimes \mathbb{C}^D$. Hence we can apply the Jordan–Chevalley decomposition

$$\overset{B^{-1}}{\underset{A}{\rule{0pt}{1pt}}} \;=\; S\rule{0pt}{1pt} \;+\; N\rule{0pt}{1pt} \;, \tag{41}$$

where $S$ is diagonalizable, $N$ is nilpotent, and $[S, N] = 0$. This decomposition always exists and is unique. It was shown in the proof of Prop. 20 of Ref. [37] that $S$ has rank one. It follows that $S$ takes the form

$$S\rule{0pt}{1pt} \;=\; \lambda \; W\!\bullet \; \bullet V \;, \tag{42}$$

for $V : \mathbb{C}^D \to \mathbb{C}^\chi$ and $W : \mathbb{C}^\chi \to \mathbb{C}^D$. Ref. [37] showed that $V$ and $W$ satisfy $VW = \mathbf{1}_{\chi \times \chi}$ and (37). Such a pair is thus a reduction from $A$ to $B$.

It is important to note that, while $S$ is determined uniquely from the decomposition of $\mathcal{T}_{AB}$, the choice of $V$ and $W$ is uniquely determined only up to a complex scalar $z \in \mathbb{C}^\times$, as we could have replaced

$$V \to z^{-1}V, \; W \to zW. \tag{43}$$

We will call $S$ a *projective reduction* since it defines a reduction up to a complex scalar as in (43). Let $PR_\chi(D)$ denote the space of projective $(\chi, D)$-reductions; this is the space $R_\chi(D)$ quotiented by the action (43).

There is an obvious projection map $R_\chi(D) \to PR_\chi(D)$, sending a reduction to its equivalence class under (43). The action is free and transitive on the fibers and the map is in fact a principal $\mathbb{C}^\times$-bundle. The bundle captures the $\mathbb{C}^\times$ ambiguity associated to choosing a reduction $V, W$ given $S$.

Now, we return to the case where we have a parametrized family of MPS. Suppose that for each $x \in U_{ab}$ we have continuous MPS tensors $A(x), B(x)$ representing the same state, with bond dimensions $D_A$ and $D_B$, respectively. Suppose that $B(x)$ is injective for each $x \in U_{ab}$. Let $D = D_A$ and $\chi = D_B$. While it is clear that the construction outlined above can be applied pointwise for $x \in U_{ab}$, we argue that the procedure can in fact be done continuously.

It is obvious that the Moore-Penrose inverse $B^{-1}(x) = (B^\dagger B)^{-1} B^\dagger$ and the tensor $\mathcal{T}_{AB}(x) = B^{-1} A$ are continuous if $A$ and $B$ are. We argue that the Jordan-Chevalley decomposition of $\mathcal{T}_{AB}(x) = S(x) + N(x)$ is also continuous when $S(x)$ is fixed to be rank one. It is generally true that $S$ and $N$ can be written as polynomials of $\mathcal{T}_{AB}$; it is then sufficient to show that the coefficients of the polynomial depend continuously on $\mathcal{T}_{AB}$.

Consider the direct sum decomposition of $\mathbb{C}^{\chi D} = \bigoplus_i V_i$ into generalized eigenspaces of $\mathcal{T}_{AB}$. By construction, $S$ acts via scalar multiplication $c_i$ on each subspace. By assumption, $S$ is rank one, so only a single $c_i = c$ is nonzero, and its corresponding eigenspace $V_i$ is one-dimensional. This fixes the form of $S$ to be

$$S = \left( \begin{array}{c|c} c & 0 \\ \hline 0 & \mathbf{0} \end{array} \right),$$

in a basis which respects the eigenspace decomposition. Here $\mathbf{0}$ denotes a $(\chi D - 1) \times (\chi D - 1)$ matrix of zeros. Nilpotency of $N$ and $[S, N] = 0$ fix $N$ to be

$$N = \left( \begin{array}{c|c} 0 & 0 \\ \hline 0 & \mathbf{J} \end{array} \right),$$

where $\mathbf{J}$ is an upper triangular and nilpotent $(\chi D - 1) \times (\chi D - 1)$ matrix; $\mathbf{J}^{\chi D - 1} = 0$. It is clear that $SN = NS = 0$. Evaluating $\mathcal{T}_{AB}^{\chi D - 1}$ gives

$$\mathcal{T}_{AB}^{\chi D - 1} = (S + N)^{\chi D - 1} = S^{\chi D - 1} = c^{\chi D - 2} S,$$

so $S = (1/c^{\chi D - 2}) \mathcal{T}_{AB}^{\chi D - 1}$. Recall that $c$ is the unique nonzero eigenvalue of $\mathcal{T}_{AB}$, which varies continuously with $\mathcal{T}_{AB}$, so $S$ is a continuous function of $\mathcal{T}_{AB}$.

As a result, given continuous $A(x)$ and a continuous injective $B(x)$, we can obtain a continuous family of projective reductions $S(x)$. In other words, we obtain a map

$$S_{AB} : U_{ab} \to PR_\chi(D), \tag{44}$$

into the space of projective reductions. Recall that the space of reductions $R_\chi(D)$ forms a $\mathbb{C}^\times$-bundle over the space of projective reductions $PR_\chi(D)$. Pulling back this $\mathbb{C}^\times$-bundle along $S_{AB}$ gives a $\mathbb{C}^\times$-bundle over $U_{ab}$.

## 5.3 Gerbe from Chern number pump

We have now developed enough machinery to describe how to construct a gerbe from the Chern number pump. This will make precise the observation made at the end of Sec. 4.

As discussed in Sec. 4, the Chern number pump is a $d = 1$ system over $S^3$ with MPS tensors $A_N^{ij}$ (27) and $A_S^{ij}$ (29) which represent the ground state and are defined on the open

sets $U_N \subset S^3$ and $U_S \subset S^3$, respectively. The tensor $A_S^{ij}$ is injective everywhere, while $A_N^{ij}$ is not injective on $U_{NS} = U_N \cap U_S$. We will construct a principal $\mathbb{C}^\times$-bundle on $U_{NS}$, which defines a gerbe on $S^3$ [42]. Since there are no triple overlaps, there are no extra data or conditions needed to specify the gerbe. This is an instance of the clutching construction.

The Moore-Penrose inverse of $A_S^{ij}$ is given by

$$(A_S^{ij})^{-1} = \langle j | \overline{U(\vec{w})} | \downarrow \rangle \langle \uparrow | U^\dagger(\vec{w}) | i \rangle \,, \tag{45}$$

which can be computed directly from (39). Calculating the tensor $\mathcal{T}_{NS} = \sum_{ij} (A_S^{ij})^{-1} A_N^{ij}$ gives

$$
\begin{aligned}
\mathcal{T}_{NS} &= \sum_{ij} \left( \langle j | \overline{U} | \downarrow \rangle \langle \uparrow | U^\dagger | i \rangle \right) \left( | i \rangle \langle j | U | \downarrow \rangle \langle \uparrow | U^T \right) \\
&= \sum_{ij} | i \rangle \langle i | \overline{U} | \uparrow \rangle \langle \downarrow | U^\dagger | j \rangle \langle j | U | \downarrow \rangle \langle \uparrow | U^T \\
&= \overline{U} | \uparrow \rangle \langle \uparrow | U^T \,,
\end{aligned} \tag{46}
$$

which reproduces the choice of reduction $V$ and $W$ used in (32). As discussed in Sec. 5.2, this defines a map

$$U_{NS} \to P\mathcal{R}_1(2) \simeq Gr_1(\mathbb{C}^2) = \mathbb{C}P^1 = S^2 \,.$$

We show in Appendix A that the space $P\mathcal{R}_1(2) \simeq Gr_1(\mathbb{C}^2)$. The canonical $\mathbb{C}^\times$-bundle over $Gr_1(\mathbb{C}^2)$ is the Hopf bundle $\mathbb{C}^2 \setminus 0 \to S^2$ with Chern number 1. The pullback yields a bundle over $U_{NS}$ with Chern number 1. The nontriviality of this bundle indicates that the gerbe over $S^3$ is nontrivial; moreover, because the Chern number is unity, the Dixmier-Douady class is a generator of $H^3(S^3, \mathbb{Z})$ [39].

The above calculation shows that the Chern number pump captures a non-torsion class in $H^3(S^3, \mathbb{Z})$. Physically, this means that stacking the system any number of times will never lead to a trivial system. As argued in Sec. 3, a non-torsion class in $H^3(S^3, \mathbb{Z})$ can never be captured by a system of MPS where the injective bond dimension is the same everywhere. This leads to the interesting conclusion that there is no way to continuously deform the family of Hamiltonians in Eq. 15 while preserving the gap such that the injective bond dimension takes the same value for all system parameters.

## 5.4  Gerbe from MPS family

We now generalize the construction to more generic families of MPS. We begin with a continuous family of states $\omega : X \to \mathcal{Q}$, where for each $x \in X$ the state $\omega(x)$ can be represented by an injective MPS tensor of bond dimension $\chi_{\omega(x)}$. Note that $\chi_{\omega(x)}$ is not generally a continuous function of $x$. We take an open cover $\{U_a\}$ of $X$ and assume the existence of continuous functions

$$
\begin{aligned}
A : U_a &\to \mathbb{C}^{nD_A^2} \,, \\
x &\mapsto A(x) \,,
\end{aligned} \tag{47}
$$

such that $A(x)$ represents the state $\omega(x)$, where $n$ is the dimension of the physical on-site Hilbert space and $D_A$ is the bond dimension of the MPS representation on $U_a$. The MPS tensor $A(x)$ is not necessarily injective. Moreover, on each double overlap $U_{ab} = U_a \cap U_b$, the restrictions of the above data give two functions $A : U_{ab} \to \mathbb{C}^{nD_A^2}$ and $B : U_{ab} \to \mathbb{C}^{nD_B^2}$. In order to compare them, we assume the existence of a continuous function $K : U_{ab} \to \mathbb{C}^{n\chi^2}$, so that $K(x)$ is injective of constant bond dimension $\chi$ and represents $\omega(x)$ for each $x \in U_{ab}$.[13] Note

---

[13]Although one might be tempted to think of this as a kind of local triviality condition on double intersections, one should resist this urge because of what we explain in Sec. 5.5 below.

that this means we assume the cover is chosen so that $\chi_{\omega(x)}$ is constant on double overlaps. The existence of the function $K$ is a property of the data $\{(U_a, A)\}$, but we do not consider $K$ itself as data; different choices of $K$ are possible and the choice will not matter.

To define a gerbe we need to construct a $\mathbb{C}^\times$-bundle on each double overlap of the cover. On $U_{ab}$, there are two tensors $A(x)$ and $B(x)$ as described above. Let us first consider the simplifying case where one of the tensors, say $B$, is injective with constant bond dimension $\chi = D_B$ on $U_{ab}$. Then, using the method described in Section 5.2, we obtain a continuous map

$$
\begin{aligned}
S_{AB} : U_{ab} &\to P\mathcal{R}_\chi(D_A), \\
x &\mapsto S_{AB}(x),
\end{aligned}
\tag{48}
$$

describing the reduction of $A(x)$ to $B(x)$ up to a complex scalar. Pulling back the $\mathbb{C}^\times$-bundle $\mathcal{R}_\chi(D_A) \to P\mathcal{R}_\chi(D_A)$ along $S_{AB}$ yields a $\mathbb{C}^\times$-bundle $\mathcal{P}_{AB}$ over $U_{ab}$. The $\mathbb{C}^\times$-bundle $\mathcal{P}_{AB}$ keeps track of the $\mathbb{C}^\times$ phase of the reduction $(V, W)$.

To proceed in the case where neither $A$ nor $B$ are injective, we first introduce the idea of a *reduction step* between two reductions. To do this we leave the setting of parametrized families for a moment and consider fixed (*i.e.* non-parametrized) MPS tensors $A$, $B$ and $K$ all representing the same state, with $K$ injective. Let $R_A = (V_A, W_A)$ and $R_B = (V_B, W_B)$ be reductions from $A$ to $K$ and $B$ to $K$ respectively. We'd like to consider all such pairs $(R_A, R_B)$, but they depend on the auxillary injective tensor $K$. We would like to get rid of this dependence. To do this, note that, given another injective tensor $K'$ representing the same state, the fundamental theorem of MPS says $K' = \lambda M^{-1} K M$ for some invertible $M \in GL(\chi)$ and $\lambda$ as in (8). From (37) it follows that changing the reference injective tensor $K \mapsto K' = \lambda M^{-1} K M$ modifies the pair $(R_A, R_B)$ as

$$
\begin{aligned}
V_A &\to M^{-1} V_A, & W_A &\to W_A M, \\
V_B &\to M^{-1} V_B, & W_B &\to W_B M.
\end{aligned}
\tag{49}
$$

To compare $A$ and $B$ when neither $A$ nor $B$ is assumed to be injective, we quotient out by this dependence. We define an equivalence relation as follows: pairs $(R_A, R_B)$ and $(R'_A, R'_B)$ are equivalent if

$$
\begin{aligned}
V'_A &= M^{-1} V_A, & W'_A &= W_A M, \\
V'_B &= M^{-1} V_B, & W'_B &= W_B M,
\end{aligned}
\tag{50}
$$

for an $M \in GL(\chi)$. A *reduction step from A to B* is an equivalence class of such pairs. We denote the equivalence class by $[R_A, R_B]$.

This motivates the definition of a space of reduction steps. Let $\chi \leq D, D'$. Consider the product

$$
\mathcal{R}_\chi(D) \times \mathcal{R}_\chi(D').
\tag{51}
$$

The *space of $(\chi, D, D')$-reduction steps*, denoted $\mathcal{R}_\chi(D, D')$, is then the quotient of $\mathcal{R}_\chi(D) \times \mathcal{R}_\chi(D')$ by the diagonal right action $(R, R') \mapsto (RM, R'M)$ for $M \in GL(\chi)$, given explicitly by

$$
((V, W), (V', W')) \mapsto ((M^{-1} V, W M), (M^{-1} V', W' M)).
\tag{52}
$$

Compare with (49). We will denote elements of $\mathcal{R}_\chi(D, D')$ by a pair $[R, R']$, with the square bracket denoting quotient by the action (52).

There is a $\mathbb{C}^\times$ action on $\mathcal{R}_\chi(D, D')$ which is given by

$$
[R, R'] \to [zR, R'],
\tag{53}
$$

where $R \to zR$ acts as in (43). This is the descendant of the $\mathbb{C}^\times$ action on the left factor of $\mathcal{R}_\chi(D) \times \mathcal{R}_\chi(D')$; the diagonal $\mathbb{C}^\times$ action is a subgroup of the diagonal $GL(\chi)$ action which was quotiented out in (50).

Let $P\mathcal{R}_\chi(D, D')$ be the quotient of $\mathcal{R}_\chi(D, D')$ by this $\mathbb{C}^\times$ action. This $\mathbb{C}^\times$ action is free and transitive on the fibers of $P\mathcal{R}_\chi(D, D')$. In fact, the quotient map $\mathcal{R}_\chi(D, D') \to P\mathcal{R}_\chi(D, D')$ is a principal $\mathbb{C}^\times$-bundle. We will denote elements of $P\mathcal{R}_\chi(D, D')$ by a pair $[\![R, R']\!]$. The double bracket denotes a quotient of $[R, R']$ by the $\mathbb{C}^\times$ action (53).

Observe that there is a continuous map

$$q\colon P\mathcal{R}_\chi(D) \times P\mathcal{R}_\chi(D') \to P\mathcal{R}_\chi(D, D'),$$
$$(S, S') \mapsto [\![R, R']\!], \tag{54}$$

where $S$ ($S'$) is the projective reduction corresponding to the reduction $R$ ($R'$). This is well-defined since the $\mathbb{C}^\times$-actions on both the right and left $\mathcal{R}_\chi(D)$ factors in (51) have been quotiented out. In particular, $[\![R, R']\!] = [\![zR, z'R']\!]$.

Now, we are ready to return to parametrized MPS. Again, suppose we have $A(x)$ and $B(x)$ defined over $U_a$ and $U_b$ respectively, with $K(x)$ injective defined on $U_{ab}$. As described above, we get continuous functions

$$S_{AK}\colon U_{ab} \to P\mathcal{R}_\chi(D_A),$$
$$S_{BK}\colon U_{ab} \to P\mathcal{R}_\chi(D_B). \tag{55}$$

These combine to give continuous maps

$$U_{ab} \to P\mathcal{R}_\chi(D_A) \times P\mathcal{R}_\chi(D_B),$$
$$x \mapsto (S_{AK}(x), S_{BK}(x)), \tag{56}$$

which we can then compose with $q$ in (54) to obtain a continuous map

$$g_{ab}\colon U_{ab} \to P\mathcal{R}_\chi(D_A, D_B),$$
$$x \mapsto [\![R_A, R_B]\!]. \tag{57}$$

This is independent of our choice of the auxilary injective tensor $K$ we used to define it. Indeed, given another continuous injective $K'$ giving rise to a map $g'_{ab}$, for each $x \in U_{ab}$ there exists $M(x) \in GL(\chi)$ and $\lambda(x) \in \mathbb{C}^\times$ with the property $K'(x) = \lambda(x)M(x)^{-1}K(x)M(x)$. The maps $g_{ab}$ and $g'_{ab}$ are equal since

$$g_{ab}(x) = [\![R_A, R_B]\!] = [\![R_A M, R_B M]\!] = [\![R'_A, R'_B]\!] = g'_{ab}(x). \tag{58}$$

This then defines a $\mathbb{C}^\times$-bundle on $U_{ab}$ by the pullback of the $\mathbb{C}^\times$-bundle

$$\mathcal{R}_\chi(D_A, D_B) \to P\mathcal{R}_\chi(D_A, D_B).$$

We denote the resulting bundle over $U_{ab}$ by $\mathcal{P}_{AB}$.

Let us make a comment on the interpretation of the bundle $\mathcal{P}_{AB}$. Recall that in the case where $B$ is injective, we obtain a reduction $R = (V, W)$ up to a phase (43). The role of the $\mathbb{C}^\times$-bundle $P_{AB}$ is that it keeps track of the phase of $W$. In the general case, we have a reduction step $[R_A, R_B]$ up to a phase (53). The role of $\mathcal{P}_{AB}$ is to keep track of the phase of the matrix $W_A V_B$. Note that the matrix $W_A V_B$ is independent of the injective reference tensor $K$ used to define it. Under the $\mathbb{C}^\times$ action (53), the matrix transforms as $W_A V_B \to z W_A V_B$; compare with the transformation of $W$ in (43).

At this stage, it is useful to see how the resulting bundle reproduces the earlier construction when one of the MPS tensors (say $B$) is injective. Picking for the moment an injective reference tensor $K$, the construction gives a projective reduction step $[\![R_A, R_B]\!]$. Since $B$ is injective, the reduction $R_B$ from $B$ to $K$ is simply a gauge transformation $V_B B W_B = K$ with $W_B = V_B^{-1}$. Following (58), we can choose an alternative reference tensor $K' = B = V_B^{-1} K V_B$, which gives

$$[\![R_A, R_B]\!] = [\![R_A V_B, R_B V_B]\!] = [\![R_{AB}, 1]\!], \tag{59}$$

where $R_{AB} = R_A V_B = (V_B^{-1} V_A, W_A V_B)$. But $R_{AB}$ is precisely the reduction from $A$ to $B$, since

$$(V_B^{-1} V_A) A (W_A V_B) = V_B^{-1} K V_B = B \, . \tag{60}$$

The projective reduction step $[\![R_{AB}, 1]\!]$ can therefore be identified with the projective reduction $S_{AB}$ in the injective case. Similarly, the reduction step $[R_A, R_B]$ can be identified with the reduction $(V, W)$ from $A$ to $B$.

Having defined a $\mathbb{C}^\times$-bundle $\mathcal{P}_{AB}$ on double overlaps $U_{ab}$, the next piece of data required to define a gerbe is the product, $\theta_{abc} : \mathcal{P}_{AB} \otimes \mathcal{P}_{BC} \to \mathcal{P}_{AC}$ over triple intersections $U_{abc}$. Roughly, the product can be described as follows. Given $[R_A, R_B]$ in $\mathcal{R}_\chi(D_A, D_B)$ and $[R'_B, R'_C]$ in $\mathcal{R}_\chi(D_B, D_C)$ which are in the same fiber so that $R'_B = z R_B$ for some $z \in \mathbb{C}^\times$, we have

$$\theta_{abc}([R_A, R_B], [R'_B, R'_C]) = [z R_A, R'_C] \, . \tag{61}$$

More formally, for any injective $K$ defined on $U_{ab}$, there is a canonical isomorphism

$$\mathcal{P}_{AK} \otimes \mathcal{P}_{BK}^{-1} \xrightarrow{\cong} \mathcal{P}_{AB} \, , \tag{62}$$

where $\mathcal{P}_{BK}^{-1}$ denotes the inverse $\mathbb{C}^\times$-bundle. It is important to note that this expression only makes sense on double intersections since $K$ is not defined on elements of the cover itself. Choose an injective tensor $K$ on the triple intersection $U_{abc}$. Then (62) holds for the restrictions of $\mathcal{P}_{AB}$, $\mathcal{P}_{BC}$ and $\mathcal{P}_{AC}$ on $U_{abc}$. The isomorphism $\theta_{abc}$ is the composite

$$\mathcal{P}_{AB} \otimes \mathcal{P}_{BC} \xrightarrow{\cong} \mathcal{P}_{AK} \otimes \mathcal{P}_{BK}^{-1} \otimes \mathcal{P}_{BK} \otimes \mathcal{P}_{CK}^{-1} \xrightarrow{\cong} \mathcal{P}_{AK} \otimes \mathcal{P}_{CK}^{-1} \xrightarrow{\cong} \mathcal{P}_{AC} \, . \tag{63}$$

Associativity is obvious from the definition of $\theta_{abc}$.

This finishes the construction of the gerbe $\mathcal{P}$ associated to a family of MPS. It follows that we get an associated cohomology class

$$d(\mathcal{P}) \in H^3(X, \mathbb{Z}) \, . \tag{64}$$

A nontrivial Dixmier-Douady class for the MPS gerbe $\mathcal{P}$ implies that it is impossible to have an MPS tensor representing the parametrized family of states which is continuous and defined everywhere on $X$. Indeed, if such a global continuous MPS tensor exists, all of the transition line bundles can be chosen to be trivial, and the resulting gerbe is trivial. This statement makes no mention of whether the MPS tensor is injective everywhere. This shows, for instance, that in the Chern number pump there is no way to extend the bond dimension $D = 2$ MPS tensor valid on the northern hemisphere to one defined globally and continuously over $S^3$. Thus, even relaxing the condition of injectivity does not allow for a global continuous MPS tensor.

We have mostly been ignoring the $H^2(X, \mathbb{Z})$ invariant discussed in Sec. 3; we now briefly comment on how to recover it for the more general MPS families discussed in this section. As explained in Sec. 3, in the case of MPS families where the injective bond dimension $\chi$ is constant over $X$, one has a principal $\mathbb{C}^\times \times PGL(\chi)$-bundle. From this one gets a $\mathbb{C}^\times$-bundle, of which the $H^2(X, \mathbb{Z})$ invariant is the first Chern class.

In the present case, we can also construct transition functions on double overlaps $h_{ab} : U_{ab} \to \mathbb{C}^\times$, from the tensors $A(x)$ and $B(x)$ defined on $U_{ab}$. In the case where $B(x)$ is injective, the projective reduction $S_{AB}$ determines a continuous $\lambda_{AB} : U_{ab} \to \mathbb{C}^\times$ by (37). We take $h_{ab} = \lambda_{AB}$. In the more general case where neither $A$ nor $B$ is necessarily injective, we have projective reductions $S_{AK}$ and $S_{BK}$ for a continuous injective tensor $K$ defined on $U_{ab}$, which give continuous maps $\lambda_{AK} : U_{ab} \to \mathbb{C}^\times$ and $\lambda_{BK} : U_{ab} \to \mathbb{C}^\times$, respectively. In this case we let $h_{ab} = \lambda_{AK} \lambda_{BK}^{-1}$. These transition functions clearly satisfy the cocycle condition on triple overlaps. Moreover, in the case of constant injective bond dimension, we recover the $\mathbb{C}^\times$-bundle discussed in Sec. 3.

## 5.5 Injective MPS families

In this section, we explore what happens if for every element $U_a$ of the cover, the MPS tensor

$$A : U_a \to \mathbb{C}^{nD_a^2}, \tag{65}$$

is injective for all $x \in U_a$. Assuming that $X$ is connected, it follows that $D_a = \chi$ independent of $a$, *i.e.* all the tensors have the same bond dimension $\chi$. We will call such families of states *injective MPS families*. As discussed in Sec. 3, injective MPS families give rise to a $PGL(\chi)$-bundle. We show that the lifting gerbe of this bundle reproduces the same invariant as our construction of a MPS gerbe.

Let us first argue that, as an intermediate step in constructing the gerbe, we reproduce the $PGL(\chi)$-bundle structure. On double overlaps $U_{ab}$, we construct projective reductions $S_{AB}$ from $A$ to $B$. The space of projective reductions is $P\mathcal{R}_\chi(\chi) = PGL(\chi)$. Since both $A$ and $B$ are injective, the reduction is in fact a gauge transformation $\lambda W_{AB}^{-1} A W_{AB} = B$, which by the fundamental theorem of MPS is unique up to a scalar multiple of $W_{AB}$. Uniqueness then implies that on triple overlaps $U_{abc}$, the MPS tensors $A$, $B$, and $C$ are related by

$$\lambda' W_{BC}^{-1} \underbrace{\lambda W_{AB}^{-1} A W_{AB}}_{B} W_{BC} = \lambda \lambda' W_{AC}^{-1} A W_{AC} = C, \tag{66}$$

so $[W_{AC}] = [W_{AB} W_{BC}]$, where the square bracket denotes quotient by an overall phase. This is precisely the cocycle condition for the $PGL(\chi)$-valued transition functions, so the injective MPS family defines a $PGL(\chi)$-bundle.

The construction of the gerbe proceeds by pulling back the bundle $GL(\chi) \to PGL(\chi)$. We then obtain line bundles $\mathcal{P}_{AB}$ on double overlaps $U_{ab}$. The product on the gerbe is given by (61); let us unpack the definition for the injective MPS family. Elements of $\mathcal{P}_{AB}$ are gauge transformations $(W_{AB}^{-1}, W_{AB})$ from $A$ to $B$; similarly elements of $\mathcal{P}_{BC}$ are gauge transformations $(W_{BC}^{-1}, W_{BC})$ from $B$ to $C$. As reduction steps, the gauge transformations are

$$\begin{aligned} W_{AB} &\to [R_{AB}, 1], \\ W_{BC} &\to [R_{BC}, 1] = [1, R_{CB}]. \end{aligned} \tag{67}$$

Applying (61) gives

$$\begin{aligned} \theta_{abc}([R_{AB}, 1], [1, R_{CB}]) &= [R_{AB}, R_{CB}] = [R_{AB} W_{BC}, 1] \\ &= [(W_{BC}^{-1} W_{AB}^{-1}, W_{AB} W_{BC}), 1] \to W_{AB} W_{BC}. \end{aligned} \tag{68}$$

In terms of the original gauge transformations, then, we have $\theta_{abc}(W_{AB}, W_{BC}) = W_{AB} W_{BC}$, which is just ordinary matrix multiplication.

We can understand the Dixmier-Douady class of this gerbe as follows. Suppose we started with a good cover so that all double intersections $U_{ab}$ are contractible. Then we can choose local sections $W_{AB} : U_{ab} \to GL(\chi)$. Then the class $g_{abc}$ is given by

$$W_{AB} W_{BC} = g_{abc} W_{AC}. \tag{69}$$

If the class is trivial, then an appropriate choice of local sections $W_{AB}$ satisfies $W_{AB} W_{BC} = W_{AC}$; this defines a lift of the $PGL(\chi)$-bundle to a $GL(\chi)$-bundle. On the other hand, a nontrivial class represents an obstruction to such a lifting. This kind of gerbe is often referred to as a lifting gerbe. Note the similarity to the physics of $d = 1$ SPTs, where SPTs with $G$-symmetry are classified by projective representations of $G$. The projective representation is a homomorphism $G \to PGL(\chi)$, and for nontrivial SPTs there is an obstruction to lifting to a homomorphism to $GL(\chi)$. [29]

The Dixmier-Douady class of the lifting gerbe of a $PGL(\chi)$-bundle is known to be torsion in $H^3(X, \mathbb{Z})$. Injective MPS families thus provide classes of parametrized systems which are invisible to the invariant obtained by integrating the higher Berry curvature, which cannot detect torsion elements of $H^3(X, \mathbb{Z})$.

# 6 Nontrivial family of injective MPS

In this section we use the suspension construction of Ref. [20] to construct a family of Hamiltonians whose ground states admit injective MPS representations with constant bond dimension $\chi = 2$. We determine that this family is nontrivial by studying the resulting $PGL(2)$-bundle, which has a nonzero Dixmier Douady class.

Consider a $d = 1$ lattice with two qubits per site, using the same notation for sites and Pauli operators as in Sec. 4. We choose the parameter space to be $X = \mathbb{R}P^2 \times S^1$. We parameterize $X$ by pairs $([\hat{n}], t)$, where $[\hat{n}]$ are equivalence classes of unit vectors in $\mathbb{R}^3$, with $\hat{n} \sim -\hat{n}$. The circle $S^1$ is parameterized by $t$ which takes values in interval $[-1, 1]$ with endpoints identified. To define the parametrized system, it will be useful to begin with a reference Hamiltonian $H([\hat{n}])$ and a family of unitaries $\mathcal{U}([\hat{n}], t)$. The family of Hamiltonians will be

$$H([\hat{n}], t) = \mathcal{U}([\hat{n}], t) H([\hat{n}]) \mathcal{U}^\dagger([\hat{n}], t). \tag{70}$$

The reference Hamiltonian is

$$H([\hat{n}]) = \sum_{i \in \mathbb{Z}} 2(\hat{n} \cdot \vec{\sigma}_{i,a})(\hat{n} \cdot \vec{\sigma}_{i,b}) - \vec{\sigma}_{i,a} \cdot \vec{\sigma}_{i,b}, \tag{71}$$

which is a sum of single-site terms and is easily seen to be gapped. Each single-site Hamiltonian specifies a gapped $d = 0$ system over $\mathbb{R}P^2$, describing the $m = 0$ state of a spin-1 particle in a Zeeman field along $\hat{n}$, where the spin-1 Hilbert space is realized as a subspace of two qubits. This $d = 0$ system was studied by Robbins and Berry [44], where they point out that the family of states over $\mathbb{R}P^2$ is nontrivial. The nontriviality lies in the $-1$ Berry phase coming from parallel transport along a noncontractible cycle in $\mathbb{R}P^2$.

Accordingly, the unique ground state of the reference Hamiltonian is a product state where each site is in the $S = 1$ state of two spin-1/2 particles with $m = 0$ along the $\hat{n}$ axis,

$$|\Psi_{[\hat{n}]}(t=0)\rangle = \bigotimes_i \left( U(\hat{n}) \otimes U(\hat{n}) \right) \left( \frac{|\uparrow\downarrow\rangle + |\downarrow\uparrow\rangle}{\sqrt{2}} \right) = \bigotimes_i \ \overset{U}{\underset{\sigma^x}{\Big|\bullet\Big|}}\overset{U}{} = \bigotimes_i \ \underset{\tilde{\sigma}^x}{\Big|\bullet\Big|}. \tag{72}$$

Here $U(\hat{n})$ is the rotation matrix given by (22). In the third equality we "pushed" $U(\hat{n})$ from the second qubit to the first and defined $\tilde{\sigma}^x = U(\hat{n})\sigma^x U^T(\hat{n})$. It can be checked that $\tilde{\sigma}^x$ is a well-defined function of $\hat{n}$, but it is *not* a well-defined function of $[\hat{n}]$. Rather, we have $\tilde{\sigma}^x \to -\tilde{\sigma}^x$ when $\hat{n} \to -\hat{n}$. This is to be expected, since the fact that the wavefunction changes sign upon $\hat{n} \to -\hat{n}$ indicates the nontriviality of the $0d$ system over $\mathbb{R}P^2$. Note that the state of the $d = 1$ system remains unchanged under $\hat{n} \to -\hat{n}$, since at most this only changes the overall phase of the wave function.

We now turn to the definition of $\mathcal{U}([\hat{n}], t)$. The full unitary takes the form

$$\mathcal{U}([\hat{n}], t) = \begin{cases} \mathcal{U}_+([\hat{n}], t), & t \geq 0, \\ \mathcal{U}_-([\hat{n}], t), & t \leq 0. \end{cases} \tag{73}$$

We build the unitaries $\mathcal{U}_+$ and $\mathcal{U}_-$ out of local gates $\mathcal{U}_{i,i+1}$ defined by

$$\mathcal{U}_{i,i+1}([\hat{n}], t) = \exp\left( i \frac{\pi}{2} t \left( 1 - (\hat{n} \cdot \vec{\sigma}_{i,b})(\hat{n} \cdot \vec{\sigma}_{i+1,a}) \right) \right). \tag{74}$$

The gate $\mathcal{U}_{i,i+1}$ interpolates between the identity at $t = 0$ and $(\hat{n} \cdot \vec{\sigma}_{i,b})(\hat{n} \cdot \vec{\sigma}_{i+1,a})$ at $t = \pm 1$. The unitaries $\mathcal{U}_+$ and $\mathcal{U}_-$ are products of $\mathcal{U}_{i,i+1}$ acting on the even and odd bonds, respectively:

$$\begin{aligned} \mathcal{U}_+([\hat{n}], t) &= \prod_{i \text{ even}} \mathcal{U}_{i,i+1}([\hat{n}], t), \\ \mathcal{U}_-([\hat{n}], t) &= \prod_{i \text{ odd}} \mathcal{U}_{i,i+1}([\hat{n}], t). \end{aligned} \tag{75}$$

Both the reference Hamiltonian and $\mathcal{U}([\hat{n}], t)$ are invariant under $\hat{n} \rightarrow -\hat{n}$, and thus are well-defined functions of $[\hat{n}]$. Moreover, it can be checked that $H([\hat{n}], t = 1) = H([\hat{n}], t = -1)$, so the full Hamiltonian $H([\hat{n}], t)$ is well-defined over $\mathbb{R}P^2 \times S^1$. While $H([\hat{n}], t)$ can be explicitly evaluated using (70), (73), (74), (75), its precise form is not very illuminating. The exception is at $t = \pm 1$, where

$$H([\hat{n}], t = \pm 1) = \sum_{i \in \mathbb{Z}} \vec{\sigma}_{i,a} \cdot \vec{\sigma}_{i,b}, \tag{76}$$

which is independent of $\hat{n}$. The family of unitaries (75) is constructed so that $H([\hat{n}], t)$ interpolates between the reference Hamiltonian (71) and (76).

The ground state over $\mathbb{R}P^2 \times S^1$ is given in terms of the $t = 0$ ground state (72) by

$$|\Psi_{[\hat{n}]}(t) = \mathcal{U}([\hat{n}], t)|\Psi_{[\hat{n}]}(t = 0)\rangle. \tag{77}$$

Because both the state and the unitaries factorize as tensor products, we have

$$|\Psi_{[\hat{n}]}(t)\rangle = \bigotimes_{i \text{ even}} \mathcal{U}_{i,i+1}([\hat{n}], t)|\psi_{[\hat{n}]}\rangle_i |\psi_{[\hat{n}]}\rangle_{i+1} = \;\;\boxed{\phantom{xx}} , \tag{78}$$

for $t \geq 0$, and

$$|\Psi_{[\hat{n}]}(t)\rangle = \bigotimes_{i \text{ odd}} \mathcal{U}_{i,i+1}([\hat{n}], t)|\psi_{[\hat{n}]}\rangle_i |\psi_{[\hat{n}]}\rangle_{i+1} = \;\;\boxed{\phantom{xx}} , \tag{79}$$

for $t \leq 0$. We have depicted the state diagrammatically with a system size of four unit cells with periodic boundary conditions; in Eq. 79 the two $\mathcal{U}$'s at the edges represent two halves of a single operator. Using $\mathcal{U}_{i,i+1}([\hat{n}], t = \pm 1) = (\hat{n} \cdot \vec{\sigma}_{i,b})(\hat{n} \cdot \vec{\sigma}_{i+1,a})$, it can be shown that at $t = \pm 1$ the state becomes

$$|\Psi_{[\hat{n}]}(t = \pm 1)\rangle = \bigotimes_i \left( \frac{|\uparrow\downarrow\rangle - |\downarrow\uparrow\rangle}{\sqrt{2}} \right)_i = \bigotimes_i \;\;\boxed{\phantom{x}}_{-i\sigma^y} , \tag{80}$$

so each site hosts a singlet state on two qubits. Note that both the Hamiltonian (76) and the state (80) at $t = \pm 1$ are independent of $[\hat{n}]$ and therefore are continuous everywhere.

The description in terms of MPS follows from the diagrammatic representation of the ground state. The Hamiltonian is translation invariant with four spins per unit cell, so the MPS tensor must have four physical indices. For a particular choice of unit cell, we "cut" the states (78) and (79) to obtain the MPS tensor,

$$A_0 = \;\;\boxed{\phantom{x}} , \tag{81}$$

for $t \geq 0$ and

$$A_0 = \;\;\boxed{\phantom{x}} , \tag{82}$$

for $t \leq 0$. Note that we have made a somewhat unusual choice of unit cell by cutting the state in between the $a$ and $b$ sublattices of a site. This choice is made so that the MPS has bond dimension $\chi = 2$ everywhere and is injective.

The MPS tensor is well-defined and continuous away from $t = \pm 1$ but suffers a discontinuity at $t = \pm 1$. In other words, it is well-defined on the open set $U_0 = \mathbb{R}P^2 \times (S^1 \setminus \pm 1)$. To see this, consider approaching $t \rightarrow 1$ from the positive side. After some algebra, $A_0$ becomes

$$A_0(t \rightarrow 1) = \;\;\boxed{\phantom{x}} . \tag{83}$$

Note that this expression depends on $[\hat{n}]$, while the state does not. On the other hand, as $t \to -1$ from the negative side, $A_0$ becomes

$$A_0(t \to -1) = \quad , \tag{84}$$

which is independent of $[\hat{n}]$. These tensors are related by a gauge transformation

$$A_0^{ijkl}(t \to -1) = (\hat{n} \cdot \vec{\sigma})^T A_0^{ijkl}(t \to 1)(\hat{n} \cdot \vec{\sigma})^T. \tag{85}$$

Note that $(\hat{n} \cdot \vec{\sigma})^T = [(\hat{n} \cdot \vec{\sigma})^T]^{-1}$, so this is a valid gauge transformation. Using this gauge transformation, one can construct a MPS tensor $A_1$ which is well-defined and continuous at $t = \pm 1$. We define $A_1$ on the region $t > 0$ to be

$$A_1^{ijkl} = (\hat{n} \cdot \vec{\sigma})^T A_0^{ijkl}(\hat{n} \cdot \vec{\sigma})^T, \tag{86}$$

such that,

$$A_1 = \quad , \tag{87}$$

while we set $A_1 = A_0$ when $t < 0$. By construction, $A_1$ is well-defined and continuous everywhere away from $t = 0$; it is an MPS tensor on the open set $U_1 = \mathbb{R}P^2 \times (S^1 \setminus 0)$.

Let us now show that this defines a nontrivial $PGL(2)$-bundle over $\mathbb{R}P^2 \times S^1$ whose lifting gerbe is nontrivial. The overlap $U_{01} = U_0 \cap U_1$ includes and is homotopy equivalent to $\mathbb{R}P^2 \times S^0$, *i.e.* two disjoint copies of $\mathbb{R}P^2$. In the region $-1 < t < 0$, the transition function is trivial since $A_1 = A_0$. In the region $0 < t < 1$, the transition function is given by the gauge transformation used to define $A_1$. Explicitly, we have

$$\begin{aligned} g_{01} : \mathbb{R}P^2 \times (0,1) &\to PGL(2), \\ ([\hat{n}], t) &\mapsto [(\hat{n} \cdot \vec{\sigma})^T], \end{aligned} \tag{88}$$

where the square bracket indicates that we should quotient by $\mathbb{C}^\times$ scalar multiples. This map defines a nontrivial $\mathbb{C}^\times$-bundle over $\mathbb{R}P^2 \times (0,1)$, which can be seen in the following way. Using the operator-state corresondence (10), we can interpret $g_{01}$ as a family of states

$$g_{01}([\hat{n}], t) = \left[(\mathbf{1} \otimes \hat{n} \cdot \vec{\sigma})|\Phi^+\rangle\right], \tag{89}$$

where the square brackets again represent a quotient by the phase. We can apply the Robbins-Berry argument [44] to this family of states to show that it is a nontrivial family over $\mathbb{R}P^2$. The Berry curvature associated to this family of states vanishes, and the wavefunction $(\mathbf{1} \otimes \hat{n} \cdot \vec{\sigma})|\Phi^+\rangle$ picks up a $-1$ phase from parallel transport along a nontrivial cycle in $\mathbb{R}P^2$ (*i.e.* a path from $\hat{n}$ to $-\hat{n}$). This implies that the family of states over $\mathbb{R}P^2$ is nontrivial, which shows that the map $g_{01}$ gives a nontrivial $\mathbb{C}^\times$-bundle. We therefore obtain a nontrivial gerbe over $X$. Since $H^3(\mathbb{R}P^2 \times S^1, \mathbb{Z}) \cong \mathbb{Z}_2$, we have constructed the gerbe with unique nontrivial Dixmier-Douady class.

## 7 Higher-dimensional generalizations

It is natural to ask whether the MPS gerbe for $d = 1$ systems can be generalized to parametrized systems in higher dimensions. Focusing on the expected $H^{d+2}(X, \mathbb{Z})$ invariant of such systems, we would like to identify a geometrical object giving rise to a class in $H^{d+2}(X, \mathbb{Z})$. A natural

option is a $d$-gerbe [33], which encompasses line bundles (0-gerbes) and gerbes (1-gerbes). Here, we outline how higher-dimensional tensor networks should lead to a $d$-gerbe. Since tensor networks in higher dimensions lack the same rigid structure of MPS, our analysis will be less rigorous than above. Rather, we will give a heuristic description of the general idea along with an explicit example for $d = 2$.

## 7.1 2-gerbe in 2-dimensional systems via PEPS

The higher-dimensional generalization of MPS is given by projected entangled pair states (PEPS). In two dimensions, a PEPS on a square lattice is defined by a 5-index tensor $A^i_{\alpha\beta\gamma\delta}$ which is then contracted into an $L \times L$ lattice as shown below for $L = 3$,

$$|\psi(A)\rangle = \qquad , \qquad (90)$$

where we take periodic boundaries, such that the open legs at the top and bottom (left and right) are contracted as indicated by the bent legs. In this diagram, and throughout this section, diagonally oriented legs will always correspond to physical degrees of freedom while vertical/horizontally oriented legs will correspond to virtual (contracted) bonds of the tensor network. By blocking the tensors of the PEPS into columns, we can get a quasi-1D MPS description of the system in terms of the *column tensor* $\mathcal{A}^{(L)}$ defined as,

$$\mathcal{A}^{(L)} = \qquad . \qquad (91)$$

Similar to the case of MPS, there is redundancy in the PEPS description of a wavefunction. However, in the case of PEPS, the form of this redundancy depends strongly on the assumptions we make about the PEPS tensor itself, and a general understanding of the redundancy without assuming any properties of the tensor is known to be unattainable [45]. Let us suppose that our PEPS has the property that the column tensor $\mathcal{A}^{(L)}$ is normal for all $L$. Then, given any other column tensor $\mathcal{B}^{(L)}$ representing the same state, Eq. 37 tells us that there are matrices $\mathcal{V}^{(L)}$ and $\mathcal{W}^{(L)}$ and a non-zero complex number $\lambda^{(L)}$ such that $\lambda^{(L)}\mathcal{V}^{(L)}\mathcal{B}^{(L)}\mathcal{W}^{(L)} = \mathcal{A}^{(L)}$ and $\mathcal{V}^{(L)}\mathcal{W}^{(L)} = \mathbf{1}$ for all $L$. Graphically,

$$= \lambda^{(L)} \quad \mathcal{V}^{(\mathcal{L})} \qquad \mathcal{W}^{(\mathcal{L})} \quad . \qquad (92)$$

The reductions $\mathcal{V}^{(L)}$ and $\mathcal{W}^{(L)}$ depend on $L$, and therefore their properties in the large-$L$ limit are not clear. However, in certain cases, which we describe shortly, the reductions can themselves be written as tensor networks,

$$\mathcal{V}^{(\mathcal{L})} \quad = \quad \begin{array}{c} V \\ V \\ V \end{array} \quad , \tag{93}$$

and similarly for $\mathcal{W}^{(L)}$, where the tensor $V$ is independent of $L$. The bond dimension of $\mathcal{V}$ is in general different from that of $A$ or $B$, so we color it red. We call this a matrix product operator (MPO) representation of the operator $\mathcal{V}^{(L)}$. Note that any MPO can be considered an MPS by simply flipping some of the legs of the tensor (see Eq. 10). We can define a normal MPO tensor as one whose corresponding MPS tensor is normal. In the cases of interest, $\mathcal{V}^{(L)}$ can be represented by an normal MPO tensor. Therefore, as in the case of MPS, there is a redundancy in the choice of normal tensors $V$ which represent $\mathcal{V}^{(L)}$.

Now, we can follow the logic used to derive the MPS gerbe in the previous sections. Suppose we have an open cover $\{U_a\}$ of $X$ and a continuous family of PEPS tensors defined on each open set. Then, at the double overlaps $U_{ab} = U_a \cap U_b$, there will be two PEPS tensors $A(x)$ and $B(x)$. We further suppose the column tensor $\mathcal{A}^{(L)}(x)$ is normal for every $L$. Then, for every $x \in U_{ab}$, we can relate the two associated column tensors $\mathcal{A}^{(L)}(x)$ and $\mathcal{B}^{(L)}(x)$ via a reduction given by a pair of operators $\mathcal{V}^{(L)}(x)$ and $\mathcal{W}^{(L)}(x)$, which can be regarded as a injective MPS. Therefore, we have a injective MPS defined at every point $x \in U_{ab}$ which, by the results of the previous sections, means there is a gerbe defined at every double overlap. This is the essential ingredient in the definition of a 2-gerbe [33], which also includes other data and conditions involving higher overlaps. This strongly suggests that a family of PEPS, subject to certain constraints on their structure which make Eqs. 92 and 93 valid, gives rise to a 2-gerbe. Topologically inequivalent 2-gerbes are classified by cohomology classes in $H^4(X, \mathbb{Z})$, which is the desired invariant.

What remains is to understand the necessary constraints on the allowed PEPS tensors, similar to how we restricted to MPS that represent the same states as injective MPS in the previous sections. This is challenging, because, in contrast to MPS, we do not have a class of PEPS that captures the kinds of systems we are interested in, while also admitting the necessary structure theorems. Nevertheless, we can consider certain subclasses of PEPS for which the desired properties of reductions can be proven rigorously, and which are suited to representing short-range entangled states. One natural route is to consider injective PEPS, which are defined similarly as injective MPS [46,47], for which Eqs. 92 and 93 do hold. However, in this case, the reduction $\mathcal{V}^{(L)}$ is a tensor product of local operators, *i.e.* an MPO of bond dimension 1. As bond dimension 1 is not sufficient to support a non-trivial gerbe, we conclude that injective PEPS are not sufficient to capture 2-gerbes. A larger class of PEPS is the semi-injective PEPS defined in Ref. [37], for which Eqs. 92 and 93 also hold, and the corresponding MPOs $\mathcal{V}^{(L)}$ can have bond dimension greater than 1. Indeed, the example we discuss in the next section has an entangled plaquette structure that is very reminiscent of semi-injective PEPS. The semi-injective PEPS are also suited to describing invertible systems that are the unique, gapped ground states of certain parent Hamiltonians with periodic boundary conditions [37]. Conversely, PEPS representing simple non-invertible phases such as quantum double models fall into the framework of $G$-injectivity [48], for which the relations between two PEPS generating the same state are more complex than Eqs. 92 and 93. Therefore, the above observations do not apply to non-invertible

systems, as expected. We leave a detailed investigation of the technical conditions needed to rigorously derive the 2-gerbe structure of PEPS to future work.

## 7.2 Two-dimensional example over $X = S^4$

We now give an explicit example of a $d = 2$ system over $S^4$ which realizes a non-trivial 2-gerbe as described above. The Hamiltonian describing this system was introduced in Ref. [20] and is defined in close analogy to the 1D model over $S^3$ defined above. We consider a two dimensional square lattice and take the parameter space to be $X = S^4$ which can be parameterized as $w = (\vec{w}, w_4, w_5)$ with $|\vec{w}|^2 + w_4^2 + w_5^2 = 1$. The family of Hamiltonians is defined by

$$H_{2d}(\vec{w}, w_4, w_5) = \sum_{i \in 2\mathbb{Z}} H_{1d}(\vec{w}, w_4) + \sum_{i \in 2\mathbb{Z}+1} \bar{H}_{1d}(\vec{w}, w_4) + \sum_{i \in 2\mathbb{Z}} H_i^{2,+}(w_5) + \sum_{i \in 2\mathbb{Z}+1} H_i^{2,-}(w_5), \quad (94)$$

where $H_{1d}$ is the $d = 1$ Hamiltonian defined in Eq. 15, and $\bar{H}_{1d}$ is obtained from $H_{1d}$ by inverting the $\vec{w}$ magnetic field term on every site. It was shown in Ref. [20] that $H_{1d}$ and $\bar{H}_{1d}$, viewed as systems over $S^3$ by fixing $|\vec{w}|^2 + w_4^2$, are inverses in the sense that they carry opposite $H^3(S^3, \mathbb{Z})$ invariants. These $d = 1$ Hamiltonians are coupled by the remaining terms depending on $w_5$,

$$
\begin{aligned}
H_i^{2,+}(w_5) &= g^+(w_5) \sum_{j \in \mathbb{Z}} \vec{\sigma}_{(i,j)} \cdot \vec{\sigma}_{(i+1,j)}, \\
H_i^{2,-}(w_5) &= g^-(w_5) \sum_{j \in \mathbb{Z}} \vec{\sigma}_{(i-1,j)} \cdot \vec{\sigma}_{(i,j)}.
\end{aligned}
\quad (95)
$$

The functions $g^\pm(w_5)$ are chosen to be

$$g^+(w_5) = \begin{cases} \sqrt{w_5^2 - 1/4}, & w_5 \geq 1/2, \\ 0, & w_5 \leq 1/2, \end{cases} \quad (96)$$

and

$$g^-(w_5) = \begin{cases} 0, & w_5 \geq -1/2, \\ \sqrt{w_5^2 - 1/4}, & w_5 \leq -1/2. \end{cases} \quad (97)$$

As before, for any value of $w_5$, at most one of $H_i^{2,+}(w_5)$ or $H_i^{2,-}(w_5)$ is nonzero. As a result, $H_{2d}(\vec{w}, w_4, w_5)$ is a sum of decoupled one-dimensional Hamiltonians acting on pairs of columns of spins, which furthermore decompose into interacting 4-spin clusters, as described in Ref. [20]. Using this, one can show that the ground state of $H_{2d}$ is gapped everywhere, and can be expressed exactly as a PEPS of finite bond dimension. We note that $H_{2d}$ is isotropic in the sense that it is invariant under 90° rotation followed by exchanging $w_4$ and $w_5$.

Let us block sites of the $d = 2$ lattice into cells consisting of pairs of columns. Then, similar to the $d = 1$ case, the entanglement between columns is either within the unit cell or between the unit cells, depending on the value of $w_5$. Suppose we choose the unit cell such that the columns within a unit cell are entangled for $w_5 < -1/2$, and columns are entangled between neighboring cells for $w_5 > 1/2$. All columns are decoupled for $-1/2 \leq w_5 \leq 1/2$. Then, we can represent the system in the region $U_S$ defined by $w_5 < 1/2$ with the following column tensor,

$$\mathcal{A}_S^{(L)} = \quad \xrightarrow{w_5 \geq -\frac{1}{2}} \quad , \quad (98)$$

which has bond dimension 1 in the horizontal direction. Therein we have introduced three new tensors. The four-legged pill-shaped tensor generates the entangled ground state of two coupled columns in of spins in $H_{2d}$. Its precise form for all values of $w$ is not important to us. What matters is that (a) it is a continuous function of $w$ and (b) when $w_5 \geq -\frac{1}{2}$, the two halves of the column decouple, as depicted. The three-legged filled circle and empty circle tensors generate the $d = 1$ states $|\psi_{1d}\rangle$ and $|\bar{\psi}_{1d}\rangle$ which are the ground states of $H_{1d}$ and $\bar{H}_{1d}$, respectively. Because of (a), $\mathcal{A}_S^{(L)}$ is a continuous function over $U_S$.

For the region $U_N$ defined by $w_5 > -1/2$, we can similarly use the following continuous column tensor,

$$\mathcal{A}_N^{(L)} = \xrightarrow{w_5 \leq \frac{1}{2}} \quad , \tag{99}$$

which has bond dimension $2^L$ in the horizontal direction. Note that we have turned some of the physical (diagonally-directed) legs into virtual (horizontally-directed) legs in order to facilitate entanglement between unit cells, similar to Eq. 27. In the overlap region $U_N \cap U_S$ (*i.e.* $-\frac{1}{2} < w_5 < \frac{1}{2}$), which is homotopy equivalent to $S^3$, we can obtain a reduction between the column tensors defined in the two hemispheres. Namely, writing $\mathcal{V}^{(L)} = \langle \psi_{1d} |$ and $\mathcal{W}^{(L)} = |\psi_{1d}\rangle$, we have,

$$\mathcal{V}^{(L)}\mathcal{A}_N^{(L)}\mathcal{W}^{(L)} = \quad = \quad = \mathcal{A}_S^{(L)}, \tag{100}$$

where the contracted columns of filled circles represent $\langle \psi_{1d} | \psi_{1d} \rangle = 1$. Therefore, the reductions between PEPS in $U_N \cap U_S$ are exactly given by the ground states of the $d = 1$ Hamiltonian from Eq. 15 which we demonstrated realize a nontrivial gerbe over $S^3$. This strongly suggests that $H_{2d}$ realizes a non-trivial 2-gerbe. Indeed, the nontrivial $H^4(X, \mathbb{Z})$ class of $H_{2d}$ was demonstrated in Ref. [20] by other means.

## 7.3 Higher dimensions

The above discussion for $d = 2$ systems suggests an inductive construction of higher gerbes for higher-dimensional systems. Suppose we have a $d$-dimensional system over $X$ and an open cover of $X$, where on each open set we have a continuous $d$-dimensional tensor network representation of the ground state. Now assume that such a parametrized family gives rise to a $d$-gerbe, as we have just argued is true for $d = 2$. Then going to $d + 1$ dimensions, under suitable assumptions, the space of reductions between two $(d+1)$-dimensional tensor network representations on double overlaps is a space of $d$-dimensional tensor networks, which by assumption defines a $d$-gerbe, thus suggesting the structure of a $(d+1)$-gerbe [33]. Inequivalent $d$-gerbes are classified by cohomology classes in $H^{d+2}(X, \mathbb{Z})$ [33], which indeed matches the "within-cohomology" part of the expected classification of $d$-dimensional invertible systems.

As mentioned above, the space of reductions between higher dimensional tensor networks is not well-understood for $d > 1$. Therefore, the above is only a heuristic indication of the kinds of structures that should be present in higher-dimensional tensor networks.

# 8 Discussion

In this paper we construct a gerbe, a generalization of a line bundle, associated to a family of $d = 1$ quantum states over $X$ which can be represented pointwise by an injective MPS tensor. To the gerbe we can associate a class $H^3(X, \mathbb{Z})$ known as the Dixmier-Douady class, which is expected to classify nontrivial parametrized systems in $d = 1$. Our work shows that this class also represents an obstruction to representing the family of states with a continuous MPS tensor defined globally over $X$. This is a natural analogue and extension of the story in $d = 0$, where the Chern class of the ground state line bundle classifies nontrivial parametrized phases and represents an obstruction to making a global continuous choice of the wave function. We also sketch the generalization to dimensions $d \geq 2$ using tensor networks, where we expect that $d$-gerbes will play a central role.

Given the results of this paper, it is interesting to ask if it is possible to construct a space of MPS-representable states $\mathcal{Q}_{MPS}$ that is a $K(\mathbb{Z}, 3)$. Based in part on the classification of symmetry protected topological phases in $d = 1$, it is believed that the space $\mathcal{Q}_{1d}$ of gapped ground states of $d = 1$ local bosonic Hamiltonians is a $K(\mathbb{Z}, 3)$ [25, 26]. Roughly speaking, we might take $\mathcal{Q}_{MPS} \subset \mathcal{Q}_{1d}$ to consist of states that can be represented by a normal MPS tensor of any finite bond dimension, *i.e.* $\mathcal{Q}_{MPS} = \cup_{\chi \in \mathbb{N}} \mathcal{M}_\chi$. All the nontrivial $d = 1$ examples of parametrized systems in this paper can be realized as maps into such a space, and we conjecture that any $d = 1$ parametrized phase has such a representative system. We note that stabilization by stacking with a fixed trivial system can be built into the construction of a space of states within the approach of Ref. [21], so we expect it should be possible to define a stable version of $\mathcal{Q}_{MPS}$ along these lines.

However, the $H^2(X, \mathbb{Z})$ invariant capturing Chern number per crystalline unit cell presents an issue. We expect this to be a phase invariant for translation-invariant systems, so one option is simply to impose translation symmetry, in which case we conjecture that $\mathcal{Q}_{MPS}$ constructed as outlined above is a $K(\mathbb{Z}, 2) \times K(\mathbb{Z}, 3)$. For systems without translation symmetry, we expect it is necessary to quotient out by decoupled $d = 0$ systems to eliminate the $H^2(X, \mathbb{Z})$ invariant,[14] but at present it is not clear how to incorporate this into the construction of a space of states.

Our understanding of the relationship between gerbes and parametrized $d = 1$ systems is by no means complete. As mentioned in Sec. 5, there are many different equivalent formulations of gerbes — examples include $PU(\mathcal{H})$-bundles, bundle gerbes, and line bundles on loop space. It would be interesting to understand whether different formulations of gerbes offer different perspectives on parametrized phases in $d = 1$.

Another important direction for future work is to formulate the notion of a (higher) connection on the MPS gerbe. One expects that the higher Berry curvature of Kapustin and Spodyneiko should correspond to the curvature of the gerbe connection. The gerbe connection would provide an appropriate notion of parallel transport and holonomy; it would be desirable for the holonomy to be a "higher Berry phase" measurable by an interference experiment. This would pave the way toward a better physical understanding of the higher Berry curvature.

Finally, the use of tensor network methods in the study of parametrized systems in $d \geq 2$ is likely to be quite fruitful. In higher dimensions, the possibility of parametrized families of topologically ordered states arises. For example, there exist nontrivial families of toric codes parametrized by $S^1$, where the $e$ and $m$ anyons are exchanged upon cycling the periodic parameter [11]. It would be interesting to understand these and related phenomena through

---

[14]Naively, one might think the $H^2(X, \mathbb{Z})$ invariant simply disappears without translation symmetry, but the situation is more subtle. For instance, consider two $d = 1$ systems over $S^2$, with Chern number 0 and 1 per unit cell, respectively. For sufficiently large finite systems with $N$ sites, the total Chern numbers will be 0 and $N$. Therefore, for arbitrarily large but finite $N$, there is no way to deform one system into the other without closing a gap somewhere on $S^2$.

the lens of tensor networks. Indeed, dualities such as the aforementioned *e-m* exchange are very naturally described within the framework of PEPS [49].

*Note added*: While this manuscript was being finalized for posting, we became aware of Ref. [50] that also studies the gerbe structure in matrix product states of one-dimensional systems. After this work was posted to arXiv and while under review, additional follow-up work discussing higher Berry connections [51,52] and generalizations to higher dimensions [53,54] appeared.

# Acknowledgments

We are grateful for useful discussions with Anton Kapustin and Norbert Schuch.

**Funding information**   This material is based upon work supported by the National Science Foundation under Grant No. DMS 2055501 (AB, MH, MP, DS). The research of MQ prior to August 2022 was supported by the NDSEG program. The work of DTS and XW was supported by the Simons Collaboration on Ultra-Quantum Matter (UQM), which is funded by grants from the Simons Foundation (651440, DTS, XW; 618615, XW). The work of MH, MQ, DTS and XW also benefited from meetings of the the UQM Simons Collaboration supported by Simons Foundation grant number 618615.

# A   The space of reductions

In this appendix we clarify some aspects of the space of reductions $\mathcal{R}_\chi(D)$ and projective reductions $P\mathcal{R}_\chi(D)$.

First recall that $V_n(\mathbb{C}^m)$ is the (non-compact) Stiefel manifold, the space of $n$-frames in $\mathbb{C}^m$, while $Gr_n(\mathbb{C}^m)$ is the Grassmanian manifold, the space of $n$-dimensional subspaces of $\mathbb{C}^m$. The map $V_n(\mathbb{C}^m) \to Gr_n(\mathbb{C}^m)$ which sends a frame to its span is a principal $GL(n)$-bundle.

In order to describe this bundle more concretely, we let $GL(n, m-n)$ be the subgroup of $GL(m)$ consisting of upper block triangular matrices

$$\begin{bmatrix} X & Z \\ 0 & Y \end{bmatrix}, \tag{A.1}$$

with $X \in GL(n)$ and $Y \in GL(m-n)$. Let $GL(I_n, m-n)$ be the subgroup of $GL(n, m-n)$ consisting of those upper triangular matrices with $X$ equal to the $n \times n$ identity matrix.

One can check that there is an isomorphism of principal $GL(n)$-bundles

$$\begin{array}{ccc}
GL(m)/GL(I_n, m-n) & \xrightarrow{\cong} & V_n(\mathbb{C}^m) \\
\downarrow & & \downarrow \\
GL(m)/GL(n, m-n) & \xrightarrow{\cong} & Gr_n(\mathbb{C}^m)
\end{array}.$$

With this in mind, the space of $(\chi, D)$-reductions, denoted by $\mathcal{R}_\chi(D)$, can be described in three equivalent ways:

1. As the space of pairs $(V, W)$ with $V : \mathbb{C}^D \to \mathbb{C}^\chi$, $W : \mathbb{C}^\chi \to \mathbb{C}^D$ such that $VW = \mathbf{1}_{\chi \times \chi}$, topologized as a subspace of $\mathcal{L}(\mathbb{C}^D, \mathbb{C}^\chi) \times \mathcal{L}(\mathbb{C}^\chi, \mathbb{C}^D)$ (where $\mathcal{L}$ denotes the space of linear maps).

2. As the space of pairs $(w, T)$ consisting of a $\chi$-frame in $\mathbb{C}^D$ and a subspace $T$ of $\mathbb{C}^D$ with the property that $\text{Span}(w) \oplus T = \mathbb{C}^D$, topologized as a subspace of $V_\chi(\mathbb{C}^D) \times Gr_{D-\chi}(\mathbb{C}^D)$.

3. As the coset space $GL(D)/(I_\chi \times GL(D - \chi))$, for $GL(D)$ endowed with its canonical topology.

Let's see why these are all the same. First, for 1. to 2., the action of $W$ on the canonical $\chi$-frame of $\mathbb{C}^\chi$ gives the $\chi$-frame $w$, and the kernel of $V$ gives the subspace $T$. Going the other direction, given $(w, T)$ as in 2., let $W$ to be the matrix which sends the canonical $\chi$-frame $(\mathbf{e}_1, \ldots, \mathbf{e}_\chi)$ of $\mathbb{C}^\chi$ to $w$, and let $V$ to be the matrix which sends the $\chi$-frame $w$ to the canonical $\chi$-frame of $\mathbb{C}^\chi$, and sends elements of $T$ to zero. To see why 2. and 3. are equivalent, consider the map which takes a matrix $M$ in $GL(D)$ with columns $m_1, \ldots, m_D$ to the pair

$$((m_1, \ldots, m_\chi), \text{Span}(m_{\chi+1}, \ldots, m_D)).$$

From these descriptions, we see that the space of $(\chi, D)$-reductions $\mathcal{R}_\chi(D)$ is homotopy equivalent to $V_\chi(\mathbb{C}^D)$; the equivalence is given by simply forgetting the subspace $T$, or by letting $Z$ go to zero in (A.1). We thus have an equivalence of principal $GL(\chi)$-bundles

$$\begin{array}{ccc} \mathcal{R}(d, D) & \xrightarrow{\;\simeq\;} & V_\chi(\mathbb{C}^D) \\ \downarrow & & \downarrow \\ GL(D)/GL(\chi) \times GL(D - \chi) & \xrightarrow{\;\simeq\;} & Gr_\chi(\mathbb{C}^D) \end{array}.$$

The (right) action of $GL(\chi)$ on $\mathcal{R}(d, D)$ is given in our three equivalent formulations, for $U \in GL(\chi)$, by

1. $(V, W) \mapsto (U^{-1} V, WU)$;

2. $(w, T) \mapsto (wU, T)$;

3. $M \mapsto M \begin{bmatrix} U & 0 \\ 0 & I_{D-\chi} \end{bmatrix} \bmod I_\chi \times GL(D - \chi).$

We can restrict the $GL(\chi)$ action on $\mathcal{R}_\chi(D)$ to the action of $\mathbb{C}^\times$ multiples of the identity. This endows the space of reductions $\mathcal{R}_\chi(D)$ with a $\mathbb{C}^\times$ action $R \mapsto zR$ given by $(V, W) \mapsto (z^{-1} V, zW)$. Compare with (43). In the language of $\chi$-frames, the action sends

$$(w_1, \ldots, w_\chi) \to (zw_1, \ldots, zw_\chi). \tag{A.2}$$

Recall that $P\mathcal{R}_\chi(D)$ the quotient of $\mathcal{R}_\chi(D)$ by the $\mathbb{C}^\times$-action (43). This quotient $P\mathcal{R}_\chi(D)$ is the space of reductions up to multiplication by a complex scalar, i.e., the projective reductions. It is homotopy equivalent to the projective Stiefel manifold $PV_\chi(\mathbb{C}^D)$, which is $V_\chi(\mathbb{C}^D)$ modulo the equivalence relation (A.2); the equivalence

$$P\mathcal{R}_\chi(D) \xrightarrow{\;\simeq\;} PV_\chi(\mathbb{C}^D), \tag{A.3}$$

is again given by forgetting the subspace $T$. We then have a diagram of principal $\mathbb{C}^\times$-bundles

$$\begin{array}{ccccc} \mathcal{R}_\chi(D) & \longrightarrow & V_\chi(\mathbb{C}^D) & \longrightarrow & V_1(\mathbb{C}^\infty) \\ \downarrow & & \downarrow & & \downarrow \\ P\mathcal{R}_\chi(D) & \xrightarrow{\;\simeq\;} & PV_\chi(\mathbb{C}^D) & \longrightarrow & PV_1(\mathbb{C}^\infty) \end{array}$$

where every commuting square is a pull-back. The bottom composite is the map

$$c_{\chi D} : P\mathcal{R}_\chi(D) \to PV_1(\mathbb{C}^\infty), \tag{A.4}$$

which sends a projective reduction $[V, W] \in P\mathcal{R}_\chi(D)$ to the span of the first column of $W$ viewed as a vector in $\mathbb{C}^\infty$ by adding zeros beyond its $D$th coordinate. The $\mathbb{C}^\times$-bundle $\mathcal{R}_\chi(D) \to P\mathcal{R}_\chi(D)$ is the pull back of the bundle $V_1(\mathbb{C}^\infty) \to PV_1(\mathbb{C}^\infty)$ along $c_{\chi D}$.

We finish by commenting on what this means for our constructions of Section 5.4 of the bundles

$$\mathcal{P}_{AB} : U_{ab} \to \mathcal{R}_\chi(D_A, D_B), \tag{A.5}$$

in the case when neither $A$ nor $B$ is injective. Recall that to construction $\mathcal{P}_{AB}$, we make use of an injective tensor $K : U_{ab} \to \mathbb{C}^{n\chi^2}$. Since pull-backs behave well under composition, the intermediate bundle $\mathcal{P}_{AK}$ can be constructed via the pull-back

$$\begin{aligned} U_{ab} &\to PV_1(\mathbb{C}^\infty), \\ x &\mapsto [w_1^A(x)], \end{aligned} \tag{A.6}$$

for $[V_A, W_A]$ a projective reduction step from $A$ to $K$ and $[w_1^A]$ the line spanned by the first column of

$$W_A = [w_1^A, \ldots, w_\chi^A].$$

A similar formula holds for $B$.

Let $V_1(\mathbb{C}^\infty) \otimes V_1(\mathbb{C}^\infty)$ be the quotient of the space $V_1(\mathbb{C}^\infty) \times V_1(\mathbb{C}^\infty)$ by the relation $(zv, w) = (v, zw)$ for $v, w \in \mathbb{C}^\infty$ and $z \in \mathbb{C}^\times$. Then

$$\begin{aligned} V_1(\mathbb{C}^\infty) \otimes V_1(\mathbb{C}^\infty) &\to PV_1(\mathbb{C}^\infty) \times PV_1(\mathbb{C}^\infty), \\ v \otimes w &\mapsto ([v], [w]), \end{aligned} \tag{A.7}$$

is the universal tensor product of $\mathbb{C}^\times$-bundles. The bundle $\mathcal{P}_{AB}$ is the pull-back of this universal tensor product bundle along

$$\begin{aligned} \mathcal{P}_{AK} \otimes \mathcal{P}_{BK}^{-1} : U_{ab} &\to PV_1(\mathbb{C}^\infty) \times PV_1(\mathbb{C}^\infty), \\ x &\mapsto ([w_1^A(x)], [\overline{w}_1^B(x)]), \end{aligned} \tag{A.8}$$

where $\overline{w}_1^B(x)$ is the complex conjugate of $w_1^B(x)$. So, in all the data carried by the reductions, only the first columns of $W_A$ and $W_B$ are necessary to construct the line bundles $\mathcal{P}_{AB}$ over $U_{ab}$. However, more was needed to describe the gerbe product, as we saw above.

# B  The space of injective MPS

We provide an elementary argument that the space of states $\mathcal{M}_\chi$ representable by injective MPS of bond dimension $\chi$ gives a model for the classifying space $B(\mathbb{C}^\times \times PGL(\chi))$ in the limit where the onsite Hilbert space dimension is allowed to be arbitrarily large.

A classifying space $BG$ for a group $G$ is the quotient of a weakly contractible space $EG$ by a proper free action of $G$. For a fixed $G$, all (CW) models of $BG$ are canonically homotopy equivalent to each other, so any method of obtaining $BG$ is as good as any other.

Let us begin by considering the space of injective MPS, which we will show is weakly contractible space with a proper free $\mathbb{C}^\times \times PGL(\chi)$ action. An injective MPS $A$ of bond dimension $\chi$ is an injective map $M_\chi(\mathbb{C}) \to \mathbb{C}^n$ given by

$$M \mapsto \sum_i \text{Tr}(A^i M^T)|i\rangle,$$

where $\{|i\rangle\}$ is a basis for the on-site Hilbert space $\mathbb{C}^n$. Denote the space of all such injective maps by $V_{\chi \times \chi}(\mathbb{C}^n)$. We can embed $\mathbb{C}^n$ in $\mathbb{C}^{n+1}$ by mapping $|i\rangle$ to itself, and this gives inclusions $V_{\chi \times \chi}(\mathbb{C}^n) \subset V_{\chi \times \chi}(\mathbb{C}^{n+1})$ and, letting $n \to \infty$, we have

$$V_{\chi \times \chi}(\mathbb{C}^\infty) := \bigcup_n V_{\chi \times \chi}(\mathbb{C}^n). \tag{B.1}$$

But for any onsite dimension $n$, identifying $M_\chi(\mathbb{C})$ with $\mathbb{C}^{\chi^2}$, we get $V_{\chi \times \chi}(\mathbb{C}^n) \cong V_{\chi^d}(\mathbb{C}^n)$, *i.e.*, the space of $\chi^2$-frames in $\mathbb{C}^n$, which is the non-compact Stiefel manifold. It is known that in the limit $n \to \infty$, this space is weakly contractible.

The $\mathbb{C}^\times \times PGL(\chi)$ action of gauge transformations translates as follows to an action on $V_{\chi \times \chi}(\mathbb{C}^\infty)$. A pair $(\lambda, N)$ in $\mathbb{C}^\times \times PGL(\chi)$ acts on an injective linear map $\phi$ via the formula

$$((\lambda, N)\phi)(M) = \lambda \phi(N^T M (N^T)^{-1}).$$

One can check that this is a proper free action on $V_{\chi \times \chi}(\mathbb{C}^\infty)$. Since the space $V_{\chi \times \chi}(\mathbb{C}^\infty)$ is weakly contractible, it qualifies as a $E(\mathbb{C}^\times \times PGL(\chi))$. Quotienting by the $\mathbb{C}^\times \times PGL(\chi)$ action yields the space $B(\mathbb{C}^\times \times PGL(\chi))$, the classifying space of $\mathbb{C}^\times \times PGL(\chi)$. The fundamental theorem of injective MPS then allows us to conclude that

$$\mathcal{M}_\chi = V_{\chi \times \chi}(\mathbb{C}^\infty)/(\mathbb{C}^\times \times PGL(\chi)) \simeq B(\mathbb{C}^\times \times PGL(\chi)) \simeq B(\mathbb{C}^\times) \times B(PGL(\chi)). \tag{B.2}$$

# C  The Brauer group

Let $X$ be a finite CW-complex. We will assume $X$ is connected for simplicity of exposition, but this is not necessary. Consider the map

$$H^3(X, \mathbb{Z}) \to H^3(X, \mathbb{R}),$$

coming from the inclusion of $\mathbb{Z}$ into $\mathbb{R}$, which maps $n$ to $2\pi n$. The kernel of this map is the torsion subgroup of $H^3(X, \mathbb{Z})$, which we will denote by $\mathrm{Tor}(H^3(X, \mathbb{Z}))$. The image is isomorphic to the quotient

$$\mathrm{Free}(H^3(X, \mathbb{Z})) := H^3(X, \mathbb{Z})/\mathrm{Tor}(H^3(X, \mathbb{Z})).$$

This is the so-called "free part" of $H^3(X, \mathbb{Z})$. We have a decomposition

$$H^3(X, \mathbb{Z}) \cong \mathrm{Tor}(H^3(X, \mathbb{Z})) \oplus \mathrm{Free}(H^3(X, \mathbb{Z})), \tag{C.1}$$

but we stress that this direct sum decomposition is not unique in general: $\mathrm{Free}(H^3(X, \mathbb{Z}))$ is not a subgroup of $H^3(X, \mathbb{Z})$ in a unique way, one has to make choices to write down (C.1).

In $H^3(X, \mathbb{R})$, the image of $\mathrm{Free}(H^3(X, \mathbb{Z}))$ is the subgroup of cohomology classes which can be represented by de Rham closed 3-forms with quantized integrals. Serre (see Theorem 1.6 of Ref. [38]) gives an identification of $\mathrm{Tor}(H^3(X, \mathbb{Z}))$ in terms that are relevant to the MPS context. We describe this result here.

Let $\mathrm{Alg}(X)$ be the set of isomorphism classes of matrix algebra bundles over $X$. That is, these are locally trivial fiber bundles whose fibers are identified with $M_\chi(\mathbb{C})$ for some $\chi \geq 1$ and whose transition functions take values in the automorphisms group of $M_\chi(\mathbb{C})$, *i.e.*, in $PGL(\chi)$. The dimension $\chi$ of the fiber must be constant on connected components, and so a matrix algebra bundle determines a principal $PGL(\chi)$-bundle on each component, and vice-versa. We have seen how to associate a class in $H^3(X, \mathbb{Z})$ to a $PGL(\chi)$-bundle, so we get a function

$$d : \mathrm{Alg}(X) \to H^3(X, \mathbb{Z}),$$

which associates to a matrix algebra bundle $E$ its Dixmier-Douady class $d(E)$. This is also called the *lifting gerbe*.

There is a special class of matrix algebra bundles, those that come from vector bundles. Given a non-zero vector bundle $V$, we can associate the matrix algebra bundle $\text{End}(V)$ whose fibers are the endomorphisms of the fibers of $V$. Over a component of $X$, these correspond precisely to those $PGL(\chi)$-bundle that can be realized as $GL(\chi)$-bundles. But, the Dixmier-Douady class of these algebra bundles is zero, *i.e.*, $d(\text{End}(V)) = 0$. So, from the perspective of gerbes, elements of $\text{Alg}(X)$ of the form $\text{End}(V)$ are trivial, and the class $d(E)$ corresponding to the lifting gerbe is the obstruction to realizing the algebra bundle $E$ as the endomorphism bundle of a vector bundle $V$.

This motivates the following construction. Define an equivalence relation on $\text{Alg}(X)$ by $E \approx F$ if there exists vector bundles $V$ and $W$ over $X$ such that

$$E \otimes \text{End}(V) \cong F \otimes \text{End}(W). \tag{C.2}$$

The Brauer group of $X$, denoted $\text{Br}(X)$, is defined to be the quotient $\text{Alg}(X)$ by the relation $\approx$.

The tensor product of matrix algebras gives $\text{Alg}(X)$ the structure of a commutative monoid. After passing to the quotient by $\approx$, one can show that this operation gives the quotient $\text{Br}(X)$ the structure of an abelian group. The Dixmier-Douady class is a morphism of monoids,

$$d(E \otimes F) = d(E) + d(F).$$

Since $d(\text{End}(V)) = 0$, $E \approx F$ implies $d(E) = d(F)$. That is, the Dixmier-Douady class induces a group homomorphism from $\text{Br}(X)$ to $H^3(X, \mathbb{Z})$.

Serre proves that, in fact, the homomorphism induced by $d$ is an isomorphism of $\text{Br}(X)$ onto the torsion subgroup $\text{Tor}(H^3(X, \mathbb{Z}))$. So, there is a decomposition

$$H^3(X, \mathbb{Z}) \cong \text{Br}(X) \oplus \text{Free}(H^3(X, \mathbb{Z})).$$

Again, the subgroup $\text{Br}(X)$ is uniquely defined, but $\text{Free}(H^3(X, \mathbb{Z}))$ depends on choices.

The relation (C.2) is very similar to stacking stabilization. Understanding precisely what it means for an injective MPS to give rise to a $GL(\chi)$-bundle, and thus an element $\text{End}(V)$ of $\text{Alg}(X)$, and what (C.2) means in terms of passage to phases will be an essential step for understanding the precise relationship between MPS states parametrized by $X$ and $H^3(X, \mathbb{Z})$.

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
