# Peer review of "Charting the space of ground states with tensor networks"

_SciPost Physics, doi:SciPost Phys. 18, 168 (2025)_

## Round 1 · Referee Report · Anonymous (Referee 1) · 2023-12-3

Report

This paper studies topological properties of families Hamiltonians defined on one-dimensional spin chains, where the family depends on some continuous parameter. A paradigmatic 0d example why this is interesting is the Chern number of a family of 0d Hamiltonians (describing bands of free fermions).

In the current work, the authors introduce a classification of such phases for families of 1d systems. A key tool are tensor network states, which are well established as tools to characterize individual 1D systems. In order to generalize this to a setting of families of 1D systems, the authors build upon and extend mathematical results from the representation theory of 1D tensor networks. From there, they can extract the necessary mathematical structure, arising from re-parametrizations of the MPS family, which allows them to construct a gerbe associated to the corresponding family. This allows them to classify the phases of the parametrized 1D system through the classification of gerbes in terms of cohomology. Finally, the authors also discuss possible generalization to higher dimensions, which, due to the lack of similarly strong statements about 2D tensor networks, are more speculative or example-based.

I think this work makes an important contribution to the classification of phases of parametrized systems using tensor networks, a topic which is both important and yet unexplored. This work is thus extremely timely, and opens up a new avenue in the field (together with Ref. 58 which appeared briefly before). It introduces new mathematical ideas which connect tensor networks and classification of phases in terms of cohomology, and should thus also form a starting point for relevant follow-up work. In addition, the paper is very well written, and reads very nicely, in particular given its heavy mathematical nature, which makes it possible to get an understanding of the results without having to dive into all details immediately.

Overall, I strongly recommend publication of the paper in its present form.

Very few minor corrections: - eq. 95: There should be no physical indices on the rhs. - pg 18, right col opposite of (95): The sentence "The semi-injective PEPS are also suited [...]" does not seem to make sense. - 6 lines above (100): Should there be a "that" between "such [that] the columns"?

  • validity: -
  • significance: -
  • originality: -
  • clarity: -
  • formatting: -
  • grammar: -

Author:  Marvin Qi  on 2025-01-14  [id 5118]

(in reply to Report 1 on 2023-12-03)

We thank the referee for their kind remarks. We have modified the manuscript to account for the corrections suggested.

---

## Round 1 · Referee Report · Anonymous (Referee 2) · 2023-12-9

Strengths

1-well written
2-detailed examples
3-interesting conclusions

Report

This paper discusses families of short-range entangled states in one dimension and beyond from the perspective of matrix product states (MPS). The paper is very nicely written, going smoothly from pedagogical discussion of the problem of classifying families, to abstract results, to detailed examples, to generalizations. I congratulate the authors on this very nice work!

I have just a few minor comments, which the authors can choose to respond to or not:

  1. This is really minor, but in equation 1, should there be a minus sign? I thought in the Zeeman effect the spin aligns with the magnetic field.

  2. A technical nitpick, in that same column you mention that "the Berry curvature can also be viewed as an element of H^2(S^2,Z)", and later you say "higher Berry curvature over X gives an invariant that is believed to be quantized and take values in H^{d+2}(X,Z)". I agree with the statement for X = S^{d+2} but I find these sentences potentially misleading. Strictly speaking, the Berry curvature can only get you the free part of Chern class, in H^2(X,R). The connection itself can give the whole Chern class however. As an example take X = RP^2 and the Berry connection to be flat but have holonomy pi over the generator of pi_1(RP^2) = Z_2. Perhaps a footnote is in order, especially since this paper focuses so much on the torsion case?

  3. Also about the classification, the identification with H^{d+2}(X,Z) likely only holds for X a sphere or in low dimensions. In general it should be E^{d+1}(X), where E is the spectrum of invertible phases mentioned later in the paper. There is a map H^{d+2}(X,Z) -> E^{d+1}(X) but it is neither injective nor surjective in general. The upshot is also that the Berry connection can occasionally have fractional Chern numbers because it lives on some twisted higher line.

  4. Below that paragraph "these developments leave open the following questions...". I would say both of those questions have had answers in the papers that you cite. Namely 1. the geometric object which plays the role of the ground state line bundle is the invertible phase itself, and 2. the obstruction is the choice of uniformly gapped boundary condition for the family (cf. boundary diabolical points in Ref 8 and anomalies in parameter space in Ref 9). This is not to say this paper doesn't add a very interesting new point of view to the discussion, but I would like the claim to have solved these problems to be tempered somewhat, and the references to the literature to be better.

  5. This is a technical question... in section 3, on page 7 you say that Ref 35 has proved A_\chi -> M_\chi is a principal C^x x PGL(\chi) bundle but only for finite system size. Why does the proof in Appendix B not generalize? Appendix B shows that the space A_\chi with onsite dimension n is the Stiefel manifold of \chi^2 frames in C^n. The quotient is M_\chi, which is a Grassmannian, and this gives us a principal C^x x PGL(\chi) bundle.

  6. By the way, based on point 5, there should be "fragile" families which are nontrivial for translation-invariant MPS with fixed on-site dimension but become trivial after embedding in a larger onsite Hilbert space.

  7. "This result strongly suggests that general nontrivial d = 1 parametrized phases cannot be described using injective MPS with a fixed bond dimension." Could we also say that the MPS family has a fixed bond dimension but the injectivity rank jumps somewhere?

  8. "It is known [45] that these lifting gerbes correspond to torsion elements of H3(X, Z)." Ref. 45 is a recent paper but I think this is known for a long time and probably some topology textbooks can be cited. The DD class in H^3(X,Z) in this case is pulled back from H^3(BPGL(\chi),Z) = Z_\chi.

  9. "In this case the associated H3(X, Z) class is trivial, and we expect the parametrized phase to be trivial, but the family of MPS is still topologically distinct from the constant family of MPS." I suppose this is analogous to a 1+1d family over S^2 where we add some doubled spin-1/2s at the boundary with Chern number over the sphere. Similar to placing a lower dimensional SPT at the boundary of another one.

  10. I quite liked the explicit example in section IV. I have just one question about it. Eqns. 18, 19 form a continuous family but not a smooth one. Should I be worried about this? It seems the bundle is definable from this parametrization but I worry that it will have a singular Berry connection.

  11. I was a little confused what to do with the indices in eqn 41. Is this understood as a map M_\chi(C) -> C^n ?

  12. I enjoyed the discussion of different injectivity conditions on PEPS. This seems like a very interesting direction for future work. I agree it is somehow intuitively clear that the MPO gauge symmetry should give the higher bundle structure in this case.

  13. Just a random notational gripe... I would advocate the term 2-line bundle instead of gerbe, since people will immediately have an idea of some kind of line bundle in their mind, and also know the category number of the construction right away. Gerbe I guess was some term from before we just started adding n- (or \infty-) in front of everything.

In conclusion, I think the paper is a very nice one, and I'm looking forward to future work from this group!

  • validity: top
  • significance: high
  • originality: high
  • clarity: top
  • formatting: perfect
  • grammar: perfect

Author:  Marvin Qi  on 2025-01-14  [id 5119]

(in reply to Report 2 on 2023-12-09)

  1. In the Zeeman effect the magnetic moment aligns with the magnetic field, but physical spin-1/2 particles can have their spin either aligned (e.g. protons) or anti-aligned (e.g. electrons or neutrons) with their magnetic moment, depending on the sign of the gyromagnetic ratio. So either sign of the Hamiltonian in (1) is physically reasonable. In any case, we chose the Hamiltonian in (1) so that the Chern number of the ground state is +1 according to the conventions of the original paper by Berry; up to positive constants our Hamiltonian is the same as that studied in Berry’s paper.

  2. We have modified the comment to indicate that only the free part of the H^{d+2}(X, Z) can be obtained by integrating the higher Berry curvature.

  3. Indeed, the correspondence with Hd+2(X, Z) should only hold for low dimensions. We have added a footnote to reflect this point.

  4. We disagree with the referee that the first question has been answered in the literature. In d = 0, the ground state line bundle is a simple geometrical object that characterizes the invertible phase. It would of course be tautological to say that the invertible phase characterizes itself. We edited this question to spell out more clearly what we mean by an object that plays a role akin to the ground state line bundle. On the second point, we agree with the referee that we could have been more specific about the kind of obstruction we are looking for, in analogy with the d = 0 case. The question now reads: ”What feature of the description of the ground state over X is obstructed when the H^{d+2}(X, Z) class is non-trivial?” To the best of our knowledge, this question has not been answered previously in the literature.

  5. We believe the proof in appendix B, as well as in the original Ref. 35, generalizes straightforwardly; we wrote it in this way because the authors of Ref. 35 proved their statements in the context of finite translation-invariant systems whose thermodynamic limit can be straightforwardly taken.

  6. This is an interesting observation; we prefer not to comment on this in the paper.

  7. Yes, this is a valid way of restating the sentence that the referee quoted. We note that the referee is not suggesting here that we change anything.

  8. We agree this has been known for a long time; we have added additional appropriate citations.

  9. We don’t understand the precise analogy between our statement and the picture sketched by the referee. However, we note that the referee is not suggesting any changes here.

  10. Because we were not concerned with defining a (higher) Berry connection, we ignored the issue of smoothness. While we expect it not to matter in this context, i.e. the gerbe data is not sensitive to this choice of how the parametrization is glued at the equator, there may be other parametrizations which are better suited for discussions of (higher) Berry connections.

  11. This is to be interpreted as a map B^{−1} : C^n → M_χ(C); we have modified the text to clarify this point.

  12. There has indeed been recent work exploring this in a certain class of tensor network in two dimensions.

  13. We tend to agree with the referee on this point of terminology but have opted to use the term “gerbe” for clarity and consistency with the literature.

---

## Round 2 · List of Changes

We list the changes made in the response to the referees.

---

## Round 3 · List of Changes

Updated journal references and included references to relevant articles that have appeared since the posting of this work to arXiv.

---

## Editorial Decision

published